# Short-chain fatty acids bind to apoptosis-associated speck-like protein to activate inflammasome complex to prevent *Salmonella* infection

Hitoshi Tsugawa[1‡]*, Yasuaki Kabe[1,2‡]*, Ayaka Kanai[1], Yuki Sugiura[1], Shigeaki Hida[3], Shun'ichiro Taniguchi[4], Toshio Takahashi[5], Hidenori Matsui[6], Zenta Yasukawa[7], Hiroyuki Itou[8], Keiyo Takubo[9], Hidekazu Suzuki[10], Kenya Honda[11], Hiroshi Handa[12], Makoto Suematsu[1]*

1 Department of Biochemistry, Keio University School of Medicine, Tokyo, Japan, 2 Japan Agency for Medical Research and Development (AMED), Core Research for Evolutional Science and Technology (CREST), Tokyo, Japan, 3 Department of Molecular and Cellular Health Sciences, Graduate School of Pharmaceutical Sciences, Nagoya City University, Nagoya, Japan, 4 Department of Comprehensive Cancer Therapy, Shinshu University School Medicine, Matsumoto, Japan, 5 Suntory Foundation for Life Sciences, Bioorganic Research Institute, Kyoto, Japan, 6 Omura Satoshi Memorial Institute, Kitasato University, Tokyo, Japan, 7 TAIYO KAGAKU CO., LTD., Mie, Japan, 8 Meiji Co., Ltd., Tokyo, Japan, 9 Department of Stem Cell Biology, Research Institute, National Center for Global Health and Medicine, Tokyo, Japan, 10 Division of Gastroenterology and Hepatology, Department of Internal Medicine, Tokai University School of Medicine, Kanagawa, Japan, 11 Department of Microbiology and Immunology, Keio University School of Medicine, Tokyo, Japan, 12 Department of Chemical Biology, Tokyo Medical University, Tokyo, Japan

‡ These authors share first authorship on this work.
* h.tsugawa@keio.jp (HT); ykabe@keio.jp (YK); gasbiology@keio.jp (MS)

**Data Availability Statement:** All data can be found in the Supporting information (S1 Data).

## Abstract

Short-chain fatty acids (SCFAs) produced by gastrointestinal microbiota regulate immune responses, but host molecular mechanisms remain unknown. Unbiased screening using SCFA-conjugated affinity nanobeads identified apoptosis-associated speck-like protein (ASC), an adaptor protein of inflammasome complex, as a noncanonical SCFA receptor besides GPRs. SCFAs promoted inflammasome activation in macrophages by binding to its ASC PYRIN domain. Activated inflammasome suppressed survival of *Salmonella enterica* serovar Typhimurium (*S.* Typhimurium) in macrophages by pyroptosis and facilitated neutrophil recruitment to promote bacterial elimination and thus inhibit systemic dissemination in the host. Administration of SCFAs or dietary fibers, which are fermented to SCFAs by gut bacteria, significantly prolonged the survival of *S.* Typhimurium–infected mice through ASC-mediated inflammasome activation. SCFAs penetrated into the inflammatory region of the infected gut mucosa to protect against infection. This study provided evidence that SCFAs suppress *Salmonella* infection via inflammasome activation, shedding new light on the therapeutic activity of dietary fiber.

**Funding:** This work was supported by AMED-CREST from the Japan Agency for Medical Research and Development, AMED (to YK, grant no.: JP17gm0710010). The *S.* Typhimurium infection model in this work was partly supported by Cross-ministerial Strategic Innovation Promotion Program (SIP), "Technologies for creating next-generation agriculture, forestry and fisheries" (funding agency: Bio-oriented Technology Research Advancement Institution, NARO) (to HT). Infrastructures for imaging mass spectrometry and metabolomics were supported in part by Ryoshoku-Kenkyukai and JST ERATO Suematsu Gas Biology Project (M.S.) until FY2015. The funders had no role in study design, data collection and analysis, decision to publish, or preparation of the manuscript.

**Competing interests:** The authors have declared that no competing interests exist.

**Abbreviations:** a.a., amino acid; ASC, apoptosis-associated speck-like protein; BMDM, bone marrow–derived macrophage; CARD, caspase activation and recruitment domain; CRISPR-Cas9, clustered regularly interspersed short palindromic repeats–CRISPR-associated protein 9; DAMP, damage-associated molecular pattern; ESI-MS, electrospray ionization–mass spectroscopy; GST, glutathione-S-transferase; HE, hematoxylin–eosin; IL, interleukin; LDH, lactate dehydrogenase; LPS, lipopolysaccharide; Lys, lysine; MCT, monocarboxylate transporter; MLN, mesenteric lymph node; NLRP, nucleotide-binding oligomerization domain-like receptor protein; PAMP, pathogen-associated molecular pattern; PCoA, principal coordinate analysis; PHGG, partially hydrolyzed guar gum; *S.* Typhimurium, *S. enterica* serovar Typhimurium; SCFA, short-chain fatty acid; SPF, specific pathogen-free; SPI-1, *Salmonella* pathogenicity island 1.

## Introduction

The gastrointestinal microbiota consists of nearly 100 trillion bacteria, comprising more than 1,000 species [1]. Gut commensal bacteria, mainly of the phyla Firmicutes and Bacteroidetes, abundantly produce short-chain fatty acids (SCFAs) by fermenting indigestible carbohydrates or dietary fiber [2]. These SCFA-producing bacteria protect against colonization by enteric pathogens, such as *Shigella* spp., *Escherichia coli*, and *Salmonella* spp. [3]. A lack of dietary fiber reportedly promotes epithelial adherence of *Citrobacter rodentium*, leading to lethal colitis in mice [4]. In addition, partially hydrolyzed guar gum (PHGG), a type of dietary fiber, prevents colonization of *Salmonella* Enteritidis in young and laying hens [5]. These reports suggest that SCFAs play a protective role against pathogenic infection in the intestinal tract.

SCFAs contain fewer than six carbons, and acetate (C2), propionate (C3), and butyrate (C4) account for 90%–95% of SCFAs in the gut [6]. The G-protein-coupled receptor GPR43 activates the release of glucagon-like peptide 1 from endocrine L cells in the intestinal tract, which leads to improvement of glucose tolerance [7]. SCFAs directly bind to GPR43 to suppress insulin-mediated fat accumulation in adipose tissue [8]. Moreover, SCFAs inhibit histone deacetylase activity [9] and thus regulate the expression of aldehyde dehydrogenases [10], the antimicrobial peptide LL-37 [11], and the transcription factor FOXP3, which promotes regulatory T-cell differentiation [12]. Further, SCFAs reportedly not only promote the production of inflammatory mediators by macrophages but also induce ROS production and phagocytosis by neutrophils [13,14]. If SCFAs strongly induce excessive ROS production by neutrophils, gut commensal bacteria controlling the colonization and/or expansion of pathogens might be eliminated [15]. The detailed effects of SCFAs on the function of immune cells with respect to protection against pathogens remain unknown. In particular, which molecule (s) in immune cells react with SCFAs to regulate innate immunity remains incompletely understood, and authentic receptors for SCFAs must be identified to reveal their protective mechanisms against pathogen infection. This study aimed to identify SCFA receptors that contribute to the regulation of innate immune responses. To this end, we used high-performance affinity nanobeads, which enable direct purification of binding proteins for small-molecule compounds [16,17]. Using SCFA-bound beads, apoptosis-associated speck-like protein containing a caspase recruiting domain (ASC) was identified as an SCFA-binding protein. ASC is a critical component of the inflammasome complex that bridges nucleotide-binding oligomerization domain-like receptor proteins (NLRPs) and caspase-1 [18,19]. ASC contains two functional domains, a PYRIN domain, which is necessary for interaction with the PYRIN domain of NLRPs, and a caspase activation and recruitment domain (CARD), which interacts with caspase-1 [20,21]. The results revealed that SCFA binding to the PYRIN domains of ASC and NLRPs triggered inflammasome complex formation to promote caspase-1 activation, resulting in interleukin (IL)-1β and IL-18 induction and protection against *Salmonella enterica* serovar Typhimurium (*S.* Typhimurium) infection.

## Results

### Identification of ASC as a novel SCFA-binding protein

We used affinity nanobeads to identify SCFA-binding proteins. As direct coupling of monocarbonates such as propionate or butyrate to beads masked their carboxylic groups, cyclic anhydrides such as succinate anhydride and glutaric anhydride were coupled to amino-modified beads (Fig 1A) to produce SCFA-conjugated beads with SCFAs containing their innate carboxylic groups. Affinity beads conjugated to acrylate, an unsaturated fatty acid, were prepared as a negative control in which malate anhydride was used. Human U937 monoblastic

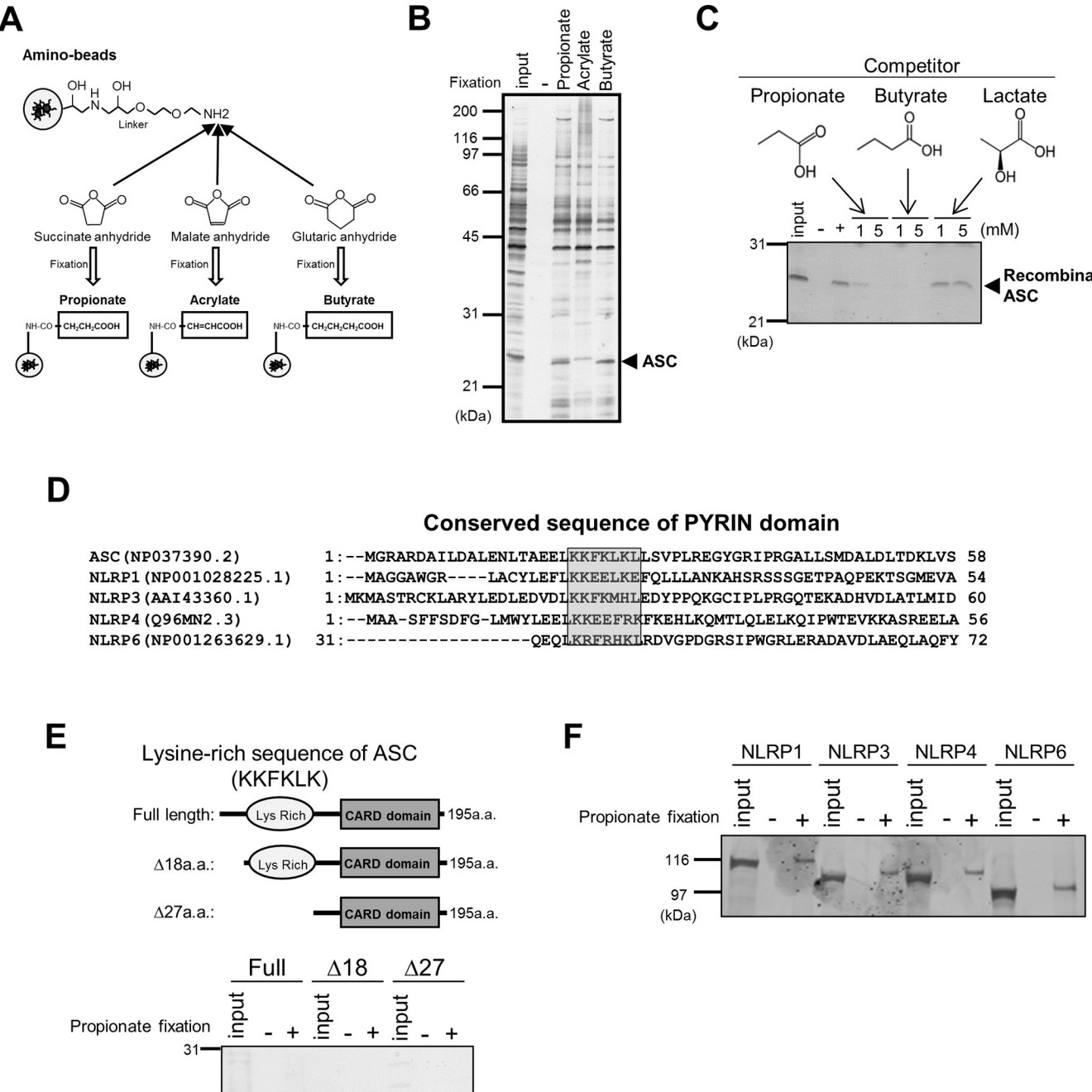

**Fig 1. SCFAs bind to PYRIN domain–containing proteins.** (A) Schematic representation of the conjugation of propionate, acrylate, and butyrate to amino-modified affinity beads using succinate anhydride, malate anhydride, and glutaric anhydride, respectively. (B) Identification of ASC/PYCARD as a SCFA-binding protein. Propionate (Pro)-, acrylate (Acryl)-, or butyrate (But)-conjugated beads were incubated with U937 cell lysate. Bound proteins were visualized by silver staining and identified by ESI-MS peptide sequencing. (C) Competition assay for binding between ASC and propionate. Recombinant ASC was pretreated with propionate, butyrate, or lactate and then incubated with control (–) or propionate-conjugated beads. Bound protein was visualized by silver staining. (D) Alignment of the a.a. sequences of the PYRIN domains of ASC and NLRP1, 3, 4, and 6 using the Genetyx program. (E) Propionate binds to the Lys-rich domain of ASC/PYCARD. Recombinant ASC (full-length or the deletion mutants) was incubated with control (–) or propionate-conjugated beads, and bound proteins were visualized by silver staining. (F) Propionate-binding assays were performed with PYRIN domain–containing NLRPs (NLRPs 1, 3, 4, and 6) using propionate-conjugated beads. a.a., amino acid; ASC, apoptosis-associated speck-like protein; CARD, caspase activation and recruitment domain;

ESI-MS, electrospray ionization–mass spectroscopy; Lys, lysine; NLRP, nucleotide-binding oligomerization domain-like receptor protein; SCFA, short-chain fatty acid.

leukemia cell line is known to exhibit the morphology and characteristics of mature macrophages and is able to respond to many exogenous stimuli. To identify authentic SCFA receptors in immune cells in order to evaluate their role in protection against pathogenic infection, we conducted a binding assay using SCFA-conjugated beads and the soluble (cytosolic) fraction of U937 cells. We detected some proteins that bind specifically to propionate and butyrate but not to acrylate (Fig 1B). Peptide sequencing by electrospray ionization (ESI)-mass spectroscopy (MS) identified a protein with an apparent molecular mass of 25 kDa as the inflammasome adaptor protein ASC, after analyzing the protein bands that specifically bound to propionate and butyrate. The specificity of ASC binding to propionate or butyrate was evaluated by a binding assay using recombinant ASC protein. ASC binding to propionate-conjugated beads was abolished by the addition of free propionate or butyrate (Fig 1C) but not by the addition of lactate, which contains a hydroxyl group. These results suggested that a linear saturated carbon backbone is necessary for ASC binding to fatty acids.

## SCFAs bind to the lysine-rich region of the PYRIN domain

The PYRIN domain is characterized by the presence of a lysine (Lys)-rich basic sequence (-KKFKLK-) that is well conserved among ASC and NLRPs and is required for binding of these proteins (Fig 1D). As SCFAs have a simple structure containing only one carboxylic acid group, we hypothesized that they bind to the basic region of the ASC PYRIN domain. To assess this hypothesis, we conducted binding assays using various deletion mutants of ASC (Fig 1E). Whereas propionate was able to bind full-length ASC and a Δ18 mutant (amino acids [a.a.] 19–195), it did not bind a Δ27 mutant (a.a. 28–195) lacking the Lys-rich sequence (Fig 1E). As NLRPs also contain the Lys-rich sequence (Fig 1D), we further examined the ability of NLRPs to bind propionate. NLRPs 1, 3, 4, and 6 also bound to the propionate-conjugated beads (Fig 1F). These results demonstrated that SCFAs can recognize the Lys-rich sequences of the PYRIN family proteins listed in Fig 1D.

## SCFAs enhance inflammasome activation by promoting the binding of ASC to NLRPs

As PYRIN domain–mediated binding of ASC to NLRPs is required for inflammasome activation, we examined the effect of SCFAs on ASC/NLRP binding by a glutathione-S-transferase (GST) pull-down assay. GST-tagged ASC protein was incubated with recombinant NLRP3 in the absence or presence of propionate, butyrate, or lactate at various concentrations (Fig 2A). Both propionate and butyrate, but not lactate, significantly enhanced the binding of NLRP3 to ASC in a dose-dependent manner. A coimmunoprecipitation assay revealed that ASC-NLRP3 interaction was also enhanced in HEK293T cells overexpressing FLAG-ASC and NLRP3 incubated with increasing concentrations of propionate or butyrate, but not in cells incubated with lactate (Fig 2B). Activation of the inflammasome complex leads to the production of mature IL-1β or IL-18 through caspase-1 activation. We examined the effect of SCFAs on IL-1β and IL-18 production in U937 cells treated with lipopolysaccharide (LPS) and ATP, which induce NLRP3-mediated inflammasome activity. The production of IL-1β and IL-18 was significantly increased upon treatment with propionate and butyrate, but not lactate (Fig 2C). IL-1β production induced by treatment with the NLRP3 activators LPS/nigericin and alum adjuvant or the NLRP1 activator anthrax lethal toxin was examined in the presence of SCFAs. Propionate

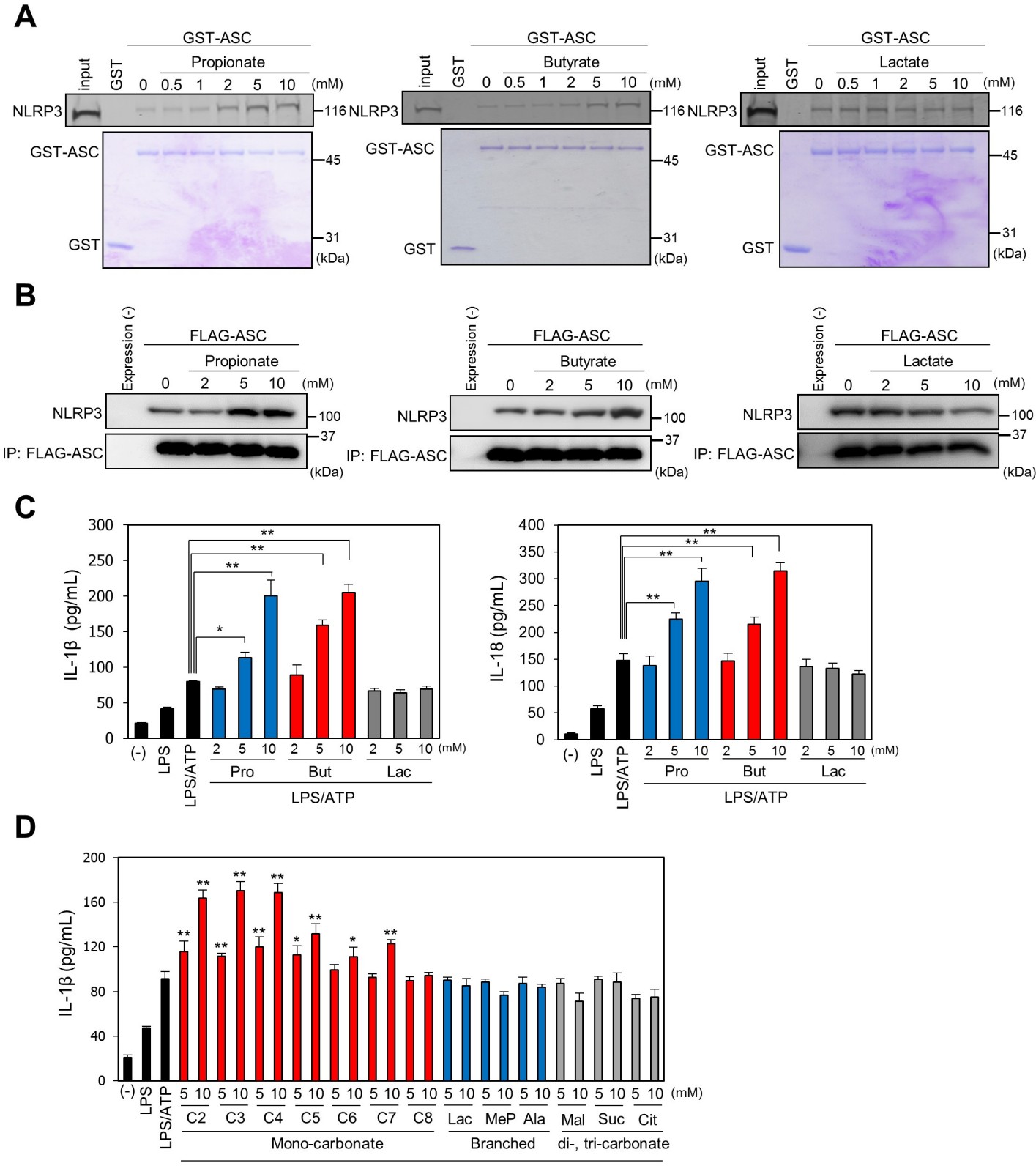

**Fig 2. SCFAs induce inflammasome activation by enhancing oligomerization of the inflammasome complex.** (A) SCFAs promote the binding between ASC and NLRP3 in vitro. GST or GST-ASC was incubated with fluorescently labeled NLRP3 in the presence or absence of propionate (left), butyrate (middle), or lactate (right).

Proteins were visualized by fluorescence imaging (top) or Coomassie Brilliant Blue staining (bottom). (B) Expression vector for FLAG-ASC and/or NLRP3 was transfected into HEK293T cells treated with propionate, butyrate, or lactate. Cell lysates were subjected to coimmunoprecipitation using anti-FLAG resin, and bound proteins were visualized by western blotting using anti-FLAG or anti-NLRP3 antibody. (C) U937 cells were incubated with LPS/ATP and treated with propionate (Pro), butyrate (But), or lactate (Lac). The IL-1β (left) and IL-18 (right) levels in the culture media were measured by ELISA. Data are the mean ± SD of three independent assays. *$P < 0.05$, **$P < 0.01$. Data are listed in S1 Data. (D) The effects of SCFA derivatives on IL-1β production were analyzed as shown above. C2, acetate; C3, propionate; C4, butyrate; C5, valerate; C6, caproate; C7, enanthate; C8, caprylate; Lac, lactate; MeP, 2-methylpropanoic acid; Ala, alanine; Mal, maleate; Suc, succinate; Cit, citrate. Data are the mean ± SD of three independent assays. *$P < 0.05$, **$P < 0.01$ versus LPS/ATP treatment. Data are listed in S1 Data. ASC, apoptosis-associated speck-like protein; GST, glutathione-S-transferase; IL, interleukin; IP, immunoprecipitation; LPS, lipopolysaccharide; NLRP, nucleotide-binding oligomerization domain-like receptor protein; SCFA, short-chain fatty acid; SD, standard deviation.

and butyrate, but not lactate, enhanced IL-1β production (S1 Fig). To clarify the structure-activity relationship between SCFAs and inflammasome activation, the effects of various fatty-acid derivatives, including mono-fatty acids, branched carboxylic acids, and di- and tri-carboxylic acids, on induced IL-1β production were analyzed (S2 Fig). LPS/ATP-induced IL-1β production was increased by treatment with mono-carboxylic fatty acids, and the activity was higher in response to fatty acids with shorter carbon chains, such as acetate (C2), propionate (C3), and butyrate (C4), than to those with longer carbon chains, such as valeric acid (C5), caproic acid (C6), enanthic acid (C7), and caprylic acid (C8). In contrast, branched and di- and tri-carboxylic acids did not affect IL-1β production (Fig 2D). These results revealed that mono-carboxylic acids with linear, saturated short chains, but not those with branched carboxylic structures, promote IL-1β production.

## SCFAs eliminate *S*. Typhimurium in macrophages by activating the inflammasome

*S*. Typhimurium invades and proliferates inside macrophages. We next examined whether SCFAs enhance the *S*. Typhimurium elimination inside murine bone marrow–derived macrophages (BMDMs) by promoting inflammasome activation. Time-course analyses revealed that upon treatment with acetate, propionate, or butyrate, but not lactate, the bacterial cell number inside BMDMs significantly decreased from 8 hours after infection and the production of IL-1β was also significantly increased compared with nontreated BMDMs, suggesting that enhanced inflammasome activation by SCFAs promote the elimination of *S*. Typhimurium in BMDMs (S3A and S3B Fig). We next examined whether the antiproliferative effect of SCFAs depends on ASC by using *ASC* knockout ($ASC^{-/-}$) BMDMs (S4A Fig). The decrease in the number of bacterial cells at 15 hours after infection detected in wild-type BMDMs treated with SCFAs was not observed in $ASC^{-/-}$ BMDMs (Fig 3A left and middle panel). As SCFAs regulate GPR43 signaling [22], we assessed the antiproliferative actions of SCFAs in *GPR43* knockout ($GPR43^{-/-}$) BMDMs infected with *S*. Typhimurium. To this end, $GPR43^{-/-}$ mice were generated by clustered regularly interspersed short palindromic repeats (CRISPR)–CRISPR-associated protein 9 (Cas9) gene editing (S4B and S4C Fig). Treatment with acetate, propionate, or butyrate, but not lactate, decreased the number of *S*. Typhimurium cells in $GPR43^{-/-}$ BMDMs (Fig 3A right panel). Treatment with acetate, propionate, or butyrate facilitated the production of cleaved caspase-1 p20, which is generated upon inflammasome activation, in wild-type and $GPR43^{-/-}$, but not $ASC^{-/-}$ BMDMs infected with *S*. Typhimurium (Fig 3B). To determine whether pyroptotic cell death is induced upon activation of caspase-1, the release of lactate dehydrogenase (LDH), an indicator of pyroptosis via caspase-1 activation, from BMDMs infected with *S*. Typhimurium was measured. LDH release from wild-type and $GPR43^{-/-}$ BMDMs infected with *S*. Typhimurium was strongly enhanced in the presence of acetate, propionate, or butyrate, but this increase was not observed in $ASC^{-/-}$ BMDMs (Figs 3C and S3C). In addition, IL-1β production in wild-type and $GPR43^{-/-}$ BMDMs infected with *S*. Typhimurium was robustly increased upon treatment with acetate, propionate, or butyrate (Fig 3D). These

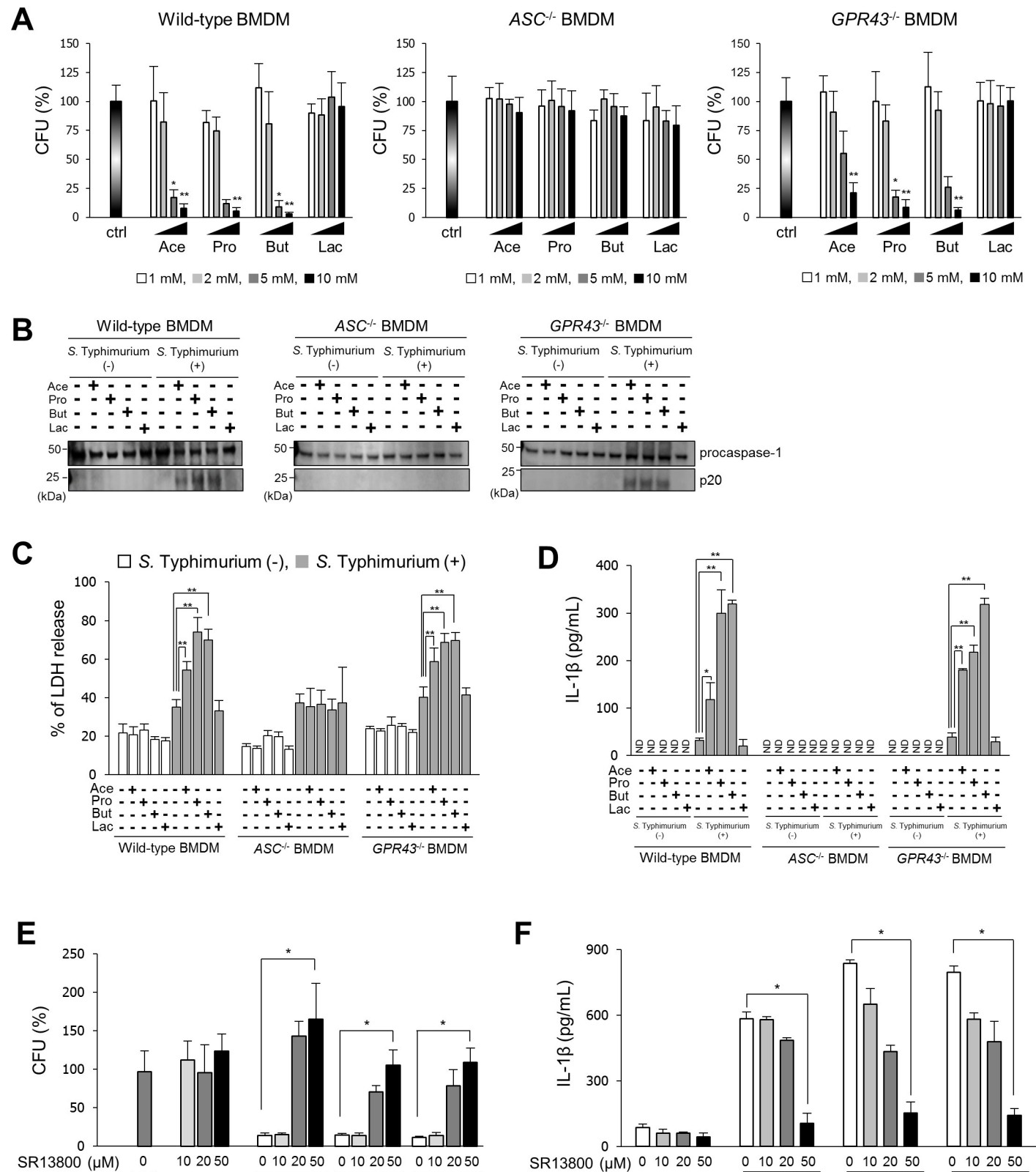

**Fig 3. SCFAs enhance the elimination of *S*. Typhimurium in macrophages by increasing inflammasome activity.** (A) BMDMs derived from wild-type, *ASC*[−/−], or *GPR43*[−/−] mice were infected with *S*. Typhimurium strain A at multiplicity of infection of 5 for 10 minutes and then incubated in DMEM containing 100 μg/mL

gentamycin for 15 hours with or without treatment with 10 mM acetate (Ace), propionate (Pro), butyrate (But), or lactate (Lac). The percentages of surviving *S.* Typhimurium strain A in comparison with untreated macrophages (ctrl) are shown. Data are the mean ± SD of three independent assays. *$P$ < 0.05, **$P$ < 0.01. Data are listed in S1 Data. (B) BMDMs were prepared as described above, and cell supernatants were collected and precipitated with 10% trichloroacetic acid. The precipitated proteins were subjected to western blotting with an anti-caspase-1 antibody. (C and D) BMDMs were prepared as described above, and cell supernatants were collected and LDH release and IL-1β production in the cell culture media as determined by LDH assay and ELISA, respectively. Data are the mean ± SD of three independent assays. *$P$ < 0.05, **$P$ < 0.01. Data listed in S1 Data. (E and F) BMDMs were treated with the MCT inhibitor SR13800 for 24 hours prior to *S.* Typhimurium strain A infection, and *S.* Typhimurium strain A–infected BMDMs were prepared as described above. The cells were lysed, and the bacterial cells within BMDMs were counted. IL-1β levels in the culture media were determined by ELISA. Data are the mean ± SD of three independent assays. *$P$ < 0.05. Data listed in S1 Data. BMDM, bone marrow–derived macrophage; CFU, colony-forming unit; DMEM, Dulbecco's modified Eagle's medium; IL, interleukin; LDH, lactate dehydrogenase; MCT, monocarboxylate transporter; ND, not detected (below the detection limit); *S.* Typhimurium, *S. enterica* serovar Typhimurium; SCFA, short-chain fatty acid; SD, standard deviation.

results showed that SCFAs accelerate bacterial elimination by enhancing inflammasome activation in *S.* Typhimurium–infected BMDMs through ASC-dependent mechanisms. As SCFA uptake is mediated by proton-linked monocarboxylate transporter (MCT) [23,24], we assessed the effect of SR13800, a potent MCT inhibitor [25], on the antiproliferative effect of SCFAs in *S.* Typhimurium–infected BMDMs. SR13800 attenuated the antiproliferative effects of SCFAs at concentrations higher than 20 μM (Fig 3E). In addition, SR13800 suppressed the increase in IL-1β production in *S.* Typhimurium–infected BMDMs induced by SCFAs, in a dose-dependent manner (Fig 3F).

Next, we investigated direct effects of SCFAs on the bacteria. Bacterial growth was not inhibited in culture medium supplemented with SCFAs (S5A Fig). As flagellin of *S.* Typhimurium activates the inflammasome [26,27], we assessed whether exposure to SCFAs would increase flagellin expression in *S.* Typhimurium. The expression of two flagellin genes (*fliC* and *fliB*) of *S.* Typhimurium was not affected by treatment with 10 mM acetate, 10 mM propionate, 10 mM butyrate, or 10 mM lactate (S5B Fig). The *Salmonella* pathogenicity island 1 (SPI-1), which is the virulence genes of *S.* Typhimurium, is also known to induce the activation of the inflammasome [28,29]. Low concentrations of SCFAs in streptomycin-treated mice enhanced colonization in the ileum by increasing SPI-1 expression, whereas SCFAs at a concentration of 30 mM decreased the expression of SPI-1 genes in vitro [30,31]. Our results showed that SCFAs at a concentration of 10 mM enhanced the elimination of *S.* Typhimurium in BMDMs by promoting inflammasome activation through ASC-dependent mechanisms. Therefore, we examined the effect of SCFAs at a concentration of 10 mM on the expression of SPI-1 genes in *S.* Typhimurium strain A. The expression of SPI-1 genes (*sipD* and *prgH*) did not change by treatment with 10 mM propionate, 10 mM butyrate, or 10 mM lactate (S6A Fig), indicating that the enhanced elimination of *S.* Typhimurium in BMDMs by 10 mM SCFAs is independent of the down-regulation of SPI-1 genes in vitro. We also evaluated the effect of SCFAs on inflammasome activation in BMDMs using *S.* Typhimurium SPI-1 knock-down strains. The SPI-1 knock-down strain of *S.* Typhimurium was constructed by introducing an in-frame deletion of *sipB*, a transcriptional activator of SPI-1, and we confirmed that the expression of SPI-1 genes (*sipD* and *prgH*) was significantly repressed in *S.* Typhimurium SPI-1 knock-down strains (S6A Fig). When using the SPI-1 knock-down strain of *S.* Typhimurium, the infection number in BMDMs and LDH release were not changed by *ASC* knock-out, and only a nonsignificant effect by SCFAs was observed in wild-type BMDMs (S6B and S6C Fig). On the other hand, the IL-1β production from wild-type BMDMs, but not *ASC*$^{−/−}$ BMDMs, infected with *S.* Typhimurium SPI-1 knock-down strains, was significantly increased by treatment with 10 mM propionate or 10 mM butyrate (S6D Fig). These results suggest that SCFAs can enhance the production of IL-1β through an ASC-dependent mechanism in macrophages infected with SPI-1 knock-down *S.* Typhimurium, but the pyroptosis associated with the elimination of the *S.* Typhimurium SPI-1 knock-down strain is independent of ASC.

## SCFAs contribute to ASC-dependent host protection against *S.* Typhimurium infection

*ASC*$^{-/-}$ mice were used to examine the involvement of ASC in host defense against *S.* Typhimurium. Specific pathogen-free (SPF) *ASC*$^{-/-}$ mice were cohoused with SPF wild-type mice and then were infected with *S.* Typhimurium. To confirm that *S.* Typhimurium strain A is not inherent pathogenicity, we first compared the pathogenicity of *S.* Typhimurium strain A with that of the *S.* Typhimurium ATCC14028S strain in mice. There was no difference in the susceptibility of mice against *S.* Typhimurium strain A and *S.* Typhimurium ATCC14028S strain (S7A Fig). In addition, in these mice, no difference was also observed in the degree of inflammatory cell infiltration in the cecal mucosa (S7B Fig). Therefore, *S.* Typhimurium strain A was used for further host protection experiments. SPF *ASC*$^{-/-}$ mice were significantly more susceptible to *S.* Typhimurium infection than wild-type SPF mice, suggesting that ASC contributes to host defense against *S.* Typhimurium infection (Fig 4A). To compare the microbiota in wild-type and *ASC*$^{-/-}$ mice, the 16S rRNA metagenomes of gut microbiota from wild-type and *ASC*$^{-/-}$ mice were analyzed. There was no significant difference in the diversity of gut microbiota between wild-type and *ASC*$^{-/-}$ mice (S8A Fig). Moreover, the results of the principal coordinate analysis (PCoA) plots also showed no difference in the microbiome composition between wild-type and *ASC*$^{-/-}$ mice (S8B Fig). These results suggest that the increased susceptibility of SPF *ASC*$^{-/-}$ mice to *S.* Typhimurium depends directly on ASC, not on a difference in the microflora. Next, to exclude the influence of SCFAs produced by gut commensal microorganisms in SPF mice, antibiotics (ampicillin, metronidazole, neomycin, and vancomycin) were orally administered to SPF mice for 4 weeks prior to *S.* Typhimurium infection. Under these conditions, there was no difference in susceptibility against *S.* Typhimurium infection between wild-type and *ASC*$^{-/-}$ mice, suggesting that the protective effect of ASC depends on the gut microbiota (Fig 4A right panel). Next, the antibiotic-treated mice were given drinking water containing 300 mM SCFAs (propionate or butyrate) or 300 mM lactate for 1 week prior to *S.* Typhimurium infection (Fig 4B). Treatment with antibiotics significantly reduced the concentrations of propionate and butyrate in the cecal lumen (S9 Fig). It was difficult to determine the exact concentration of the SCFAs of the cecal lumen of the mice given the drinking water containing 300 mM SCFAs because the timing and amount of water intake are different in each mouse, resulting in inconsistent data. Therefore, to measure the concentration of SCFAs in the cecal lumen of the mice immediately after exogenous administration of SCFAs, the mice were forced to drink orally with 100 μL of 300 mM SCFAs before 2 hours, and the SCFA concentrations in the cecal lumen were measured. The exogenous administration of SCFAs restored their concentrations to the levels observed in SPF mice not treated with antibiotics (S9 Fig). The lactate concentration in the cecal lumen was not substantially decreased upon antibiotics administration and was significantly increased upon oral administration of lactate (S9 Fig). Administration of propionate or butyrate, but not lactate, significantly prolonged the survival of *S.* Typhimurium–infected wild-type mice (Fig 4C). Notably, propionate- and butyrate-treated *GPR43*$^{-/-}$ mice showed significantly improved resistance to *S.* Typhimurium infection (S10 Fig), suggesting that the effect of exogenous SCFAs is not mediated by GPR43. In contrast, *S.* Typhimurium–infected *ASC*$^{-/-}$ mice did not show any improvement in survival upon treatment with propionate or butyrate (Fig 4D). These findings demonstrated that propionate and butyrate are essential for ASC-dependent host-protective responses. We next examined the bacterial burden in the cecum contents, cecum tissues, mesenteric lymph nodes (MLNs), livers, and spleens of infected mice at 2 days post infection. We observed no significant change in the number of bacterial cells in the cecum contents upon administration of propionate or butyrate, suggesting that SCFAs do not promote bacterial evasion from the

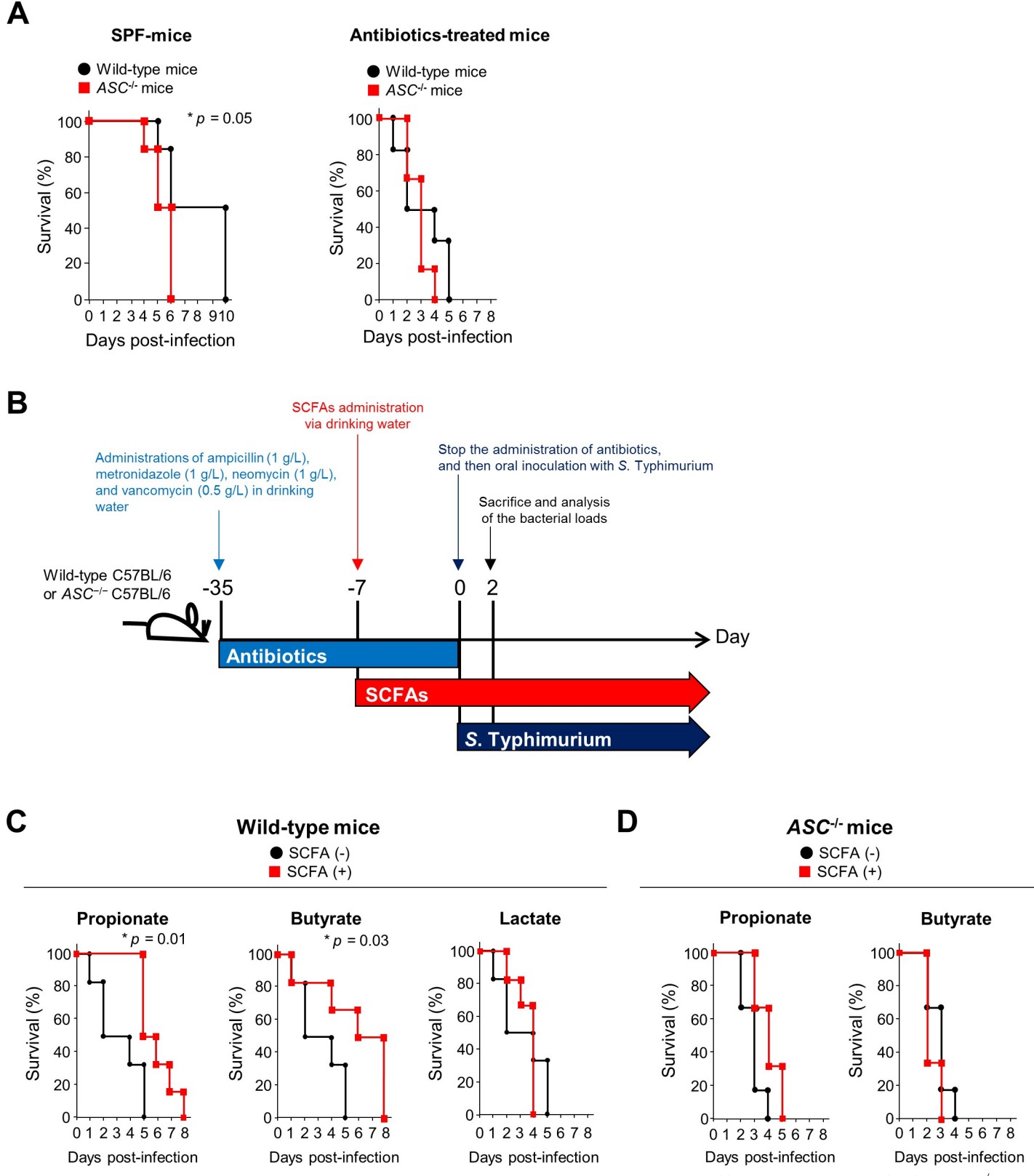

**Fig 4. SCFAs enhance host defense against *S.* Typhimurium infection in an ASC-dependent manner.** (A) SPF wild-type mice were cohoused with SPF $ASC^{-/-}$ mice at a 1:1 gender ratio for 7 days and then were infected with *S.* Typhimurium strain A. Survival of SPF wild-type (●) and SPF $ASC^{-/-}$ mice (■) infected with *S.* Typhimurium strain A (left panel). Survival of antibiotic-treated wild-type (●) and $ASC^{-/-}$ mice (■) infected with *S.* Typhimurium strain A (right panel). *P*-values were

determined by the log-rank test (*n* = 6 per group). Data are listed in S1 Data. (B) Treatment scheme followed to analyze the effect of SCFA administration on the susceptibility of antibiotic-treated mice to *S.* Typhimurium strain A. SPF mice were given antibiotics for 4 weeks. SCFAs were given 1 week before *S.* Typhimurium strain A infection. The administration of antibiotics was stopped before *S.* Typhimurium strain A infection. (C and D) Effects of propionate, butyrate, and lactate on the survival of wild-type (C) and *ASC*$^{-/-}$ (D) mice infected with *S.* Typhimurium strain A. The mice were given antibiotics for 1 month prior to being given drinking water containing 300 mM propionate, 300 mM butyrate, or 300 mM lactate for 1 week. Then, mice were inoculated with *S.* Typhimurium ($10^8$ bacteria) orally. *P*-values were determined by the log-rank test (*n* = 6 per group). Data are listed in S1 Data. ASC, apoptosis-associated speck-like protein; *S.* Typhimurium, *S. enterica* serovar Typhimurium; SCFA, short-chain fatty acid; SPF, specific pathogen-free.

gut lumen (S11A Fig). In contrast, bacterial loads in the cecum tissues, MLN, liver, and spleen in wild-type mice were significantly reduced upon administration of propionate or butyrate (S11A Fig). These effects of SCFAs were not observed in *ASC*$^{-/-}$ mice (S11B Fig).

## SCFAs recruit neutrophils into *S.* Typhimurium–infected cecal mucosa via an ASC-dependent mechanism

Acute inflammatory cell infiltration plays a critical role in innate immunity against invading pathogens [32,33]. Therefore, we examined the effects of SCFAs on the local population of inflammatory cells in the cecal mucosa of *S.* Typhimurium–infected mice. As shown in Fig 5A, *S.* Typhimurium infection induced inflammatory cell infiltration into the cecal mucosa in SPF wild-type mice. In contrast, in SPF *ASC*$^{-/-}$ mice infected with *S.* Typhimurium, inflammatory cell infiltration was not evident, indicating that ASC plays a critical role in inflammatory cell infiltration of the cecum (Fig 5A). Moreover, inflammatory cell infiltration was not induced by *S.* Typhimurium infection in antibiotic-treated wild-type mice (Fig 5B). Administration of propionate or butyrate induced inflammatory cell infiltration in response to *S.* Typhimurium infection in antibiotic-treated wild-type mice, but not *ASC*$^{-/-}$ mice (Fig 5B). The inflammatory infiltrate score of the hematoxylin–eosin (HE)-stained tissues was calculated according to a previously published method [34]. The inflammatory infiltrate scores in *S.* Typhimurium–infected wild-type mice given propionate or butyrate were significantly higher than in mice that had not been treated with SCFAs or in *S.* Typhimurium–infected *ASC*$^{-/-}$ mice (S12A Fig). To determine whether administration of SCFAs to SPF mice without *S.* Typhimurium infection induces inflammatory cell infiltration, cecum tissues from SCFA-treated SPF wild-type or *ASC*$^{-/-}$ mice without *S.* Typhimurium infection were analyzed. Inflammatory cell infiltration was not detected in the cecal mucosa or submucosa of either mouse strain, indicating that SCFAs do not induce cecal inflammation without notable pathogenic bacterial infection (S13 Fig). Propionate and butyrate induced the migration of macrophages (F4/80-positive cells) into the cecal mucosa and submucosa of wild-type mice infected with *S.* Typhimurium; in contrast, these effects were not observed in *ASC*$^{-/-}$ mice infected with *S.* Typhimurium (Fig 5C). The staining intensities of F4/80 in the cecal mucosa and submucosa of *S.* Typhimurium–infected wild-type mice given propionate or butyrate were significantly higher than in *S.* Typhimurium–infected wild-type mice without SCFA treatment or in *S.* Typhimurium–infected *ASC*$^{-/-}$ mice (S12B left panel Fig). Propionate and butyrate significantly enhanced IL-1β production in antibiotic-treated wild-type mice, but not *ASC*$^{-/-}$ mice (Fig 5D). Neutrophils, which are recruited by IL-1β, are critical innate immune cells for protecting the host against pathogens [32,33,35]. As shown in Fig 5C, propionate and butyrate induced the infiltration of neutrophils (Gr-1-positive cells) into the cecal mucosa and submucosa of wild-type, but not into those of *ASC*$^{-/-}$ mice infected with *S.* Typhimurium. The staining intensities of Gr-1 in the cecal mucosa and submucosa of *S.* Typhimurium–infected wild-type mice that were orally given propionate or butyrate were significantly higher than in *S.* Typhimurium–infected wild-type mice without SCFA treatment or in *S.* Typhimurium–infected *ASC*$^{-/-}$ mice (S12B right panel Fig). In agreement with the immunostaining results, the neutrophil marker Gr-1 was

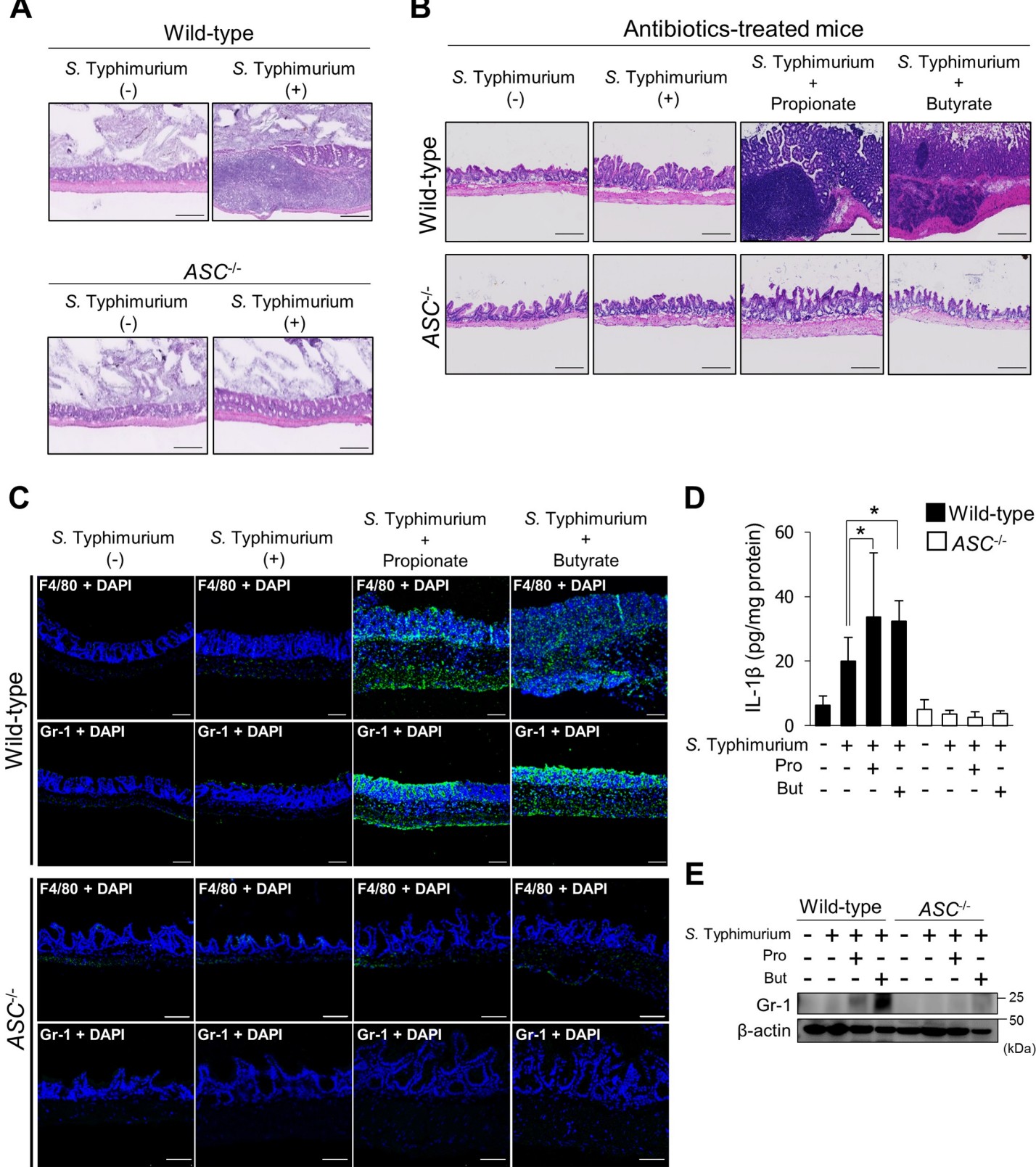

**Fig 5. SCFAs recruit neutrophils to *S.* Typhimurium–infected foci through ASC-dependent inflammasome activation.** (A and B) HE staining of cecum tissues from SPF wild-type or *ASC*$^{-/-}$ mice and antibiotic-treated wild-type and *ASC*$^{-/-}$ mice infected with *S.* Typhimurium strain A. Scale bars = 10 μm. (C) Cecal mucosa was

immunostained with an anti-F4/80 or anti-Gr-1 antibody after infection with *S*. Typhimurium strain A. Scale bars = 100 μm. (D) IL-1β production in the cecum tissues was measured by ELISA. Data are the mean ± SD. *$P < 0.05$. Data are listed in S1 Data. (E) Detection of the neutrophil marker Gr-1 protein by western blotting. Cecum tissues were collected, homogenized, and analyzed by western blotting using anti-Gr-1 and anti-actin antibodies. ASC, apoptosis-associated speck-like protein; HE, hematoxylin–eosin; IL, interleukin; *S*. Typhimurium, *S. enterica* serovar Typhimurium; SCFA, short-chain fatty acid; SD, standard deviation; SPF, specific pathogen-free.

detected in cecal tissues of wild-type mice that were given propionate or butyrate, but not in those of $ASC^{-/-}$ mice (Fig 5E). These findings suggested that SCFAs induce neutrophil recruitment into the *S*. Typhimurium–infected cecal mucosa through ASC-mediated inflammasome activation. Whereas excessive ROS production by neutrophils might eliminate gut commensal bacteria that prevent the colonization and/or expansion of *S*. Typhimurium [15], rapid neutrophil infiltration efficiently prevents the spread of the pathogen [36]. Our results suggest that SCFAs promote the elimination of *S*. Typhimurium by efficiently promoting neutrophil infiltration of the cecal mucosa by enhancing inflammasome activation in macrophages.

## Macrophage activation is required for the protective effect of SCFAs against *S*. Typhimurium infection

To investigate the potential role of macrophages in the ASC-dependent host-protective effect of SCFAs, we administrated clodronate-encapsulated liposomes, which deplete macrophages in vivo, to SCFA-treated mice for 24 hours prior to *S*. Typhimurium infection. As shown in Fig 6A, the resistance against *S*. Typhimurium infection in SCFA-treated mice was significantly reduced upon clodronate-encapsulated liposome administration. Moreover, clodronate-encapsulated liposome administration abolished the SCFA-induced decrease in bacterial cell numbers in cecum tissues, MLN, liver, and spleen (Fig 6B). Under these circumstances, infiltration of macrophages (F4/80-positive cells) and neutrophils (Gr-1-positive cells) into *S*. Typhimurium–infected cecal mucosa were not detected (Fig 6C). In addition, enhanced IL-1β production induced by SCFAs was not observed in *S*. Typhimurium–infected cecal mucosa (Fig 6D). These findings suggested that SCFAs induce neutrophil recruitment into *S*. Typhimurium–infected cecal mucosa through ASC-mediated activation of the inflammasome of macrophages, and this potentiates pathogen elimination in the infected mucosa. To assess the localization of *S*. Typhimurium in the cecal tissues of mice administrated with clodronate-encapsulated liposomes, the cecal mucosa was immunostained for *S*. Typhimurium. Whereas colocalization of *S*. Typhimurium and infiltrating macrophages (F4/80-positive cells) was detected in the cecal mucosa of SPF mice that had not been administered clodronate-encapsulated liposomes, increased staining of *S*. Typhimurium in the extracellular space was observed in the cecal submucosa of SPF mice treated with clodronate-encapsulated liposomes (S14 Fig). This result indicates that *S*. Typhimurium can invade and replicate extracellularly in the cecal submucosa of mice without macrophages. *S*. Typhimurium is translocated from the gastrointestinal tract to the bloodstream by infected phagocytes including macrophages [37,38]. Our findings suggested that SCFAs enhance bacterial elimination in the cecal mucosa by inducing neutrophil infiltration through mechanisms involving ASC-activating macrophages and thereby restrict the systemic spreading of *S*. Typhimurium. To confirm this possibility, we examined whether SCFAs activate the macrophages in MLN and Peyer's patches. IL-1β production in MLN and Peyer's patches was not significantly enhanced, indicating that SCFAs do not promote bacterial killing in MLN and Peyer's patches via an ASC-dependent mechanism (S15 Fig). These findings indicated that SCFA-induced ASC-dependent activation of macrophages in the cecal mucosa promotes bacterial elimination to limit bacterial dissemination from the gastrointestinal tract to systemic sites.

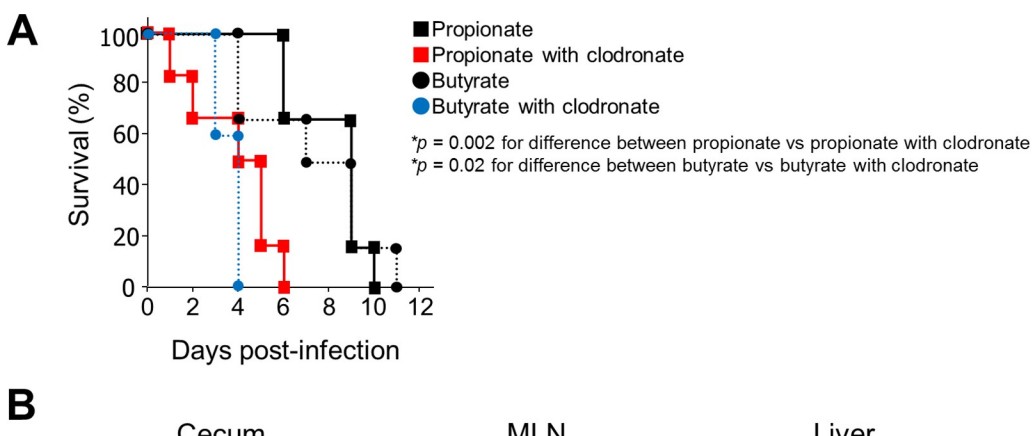

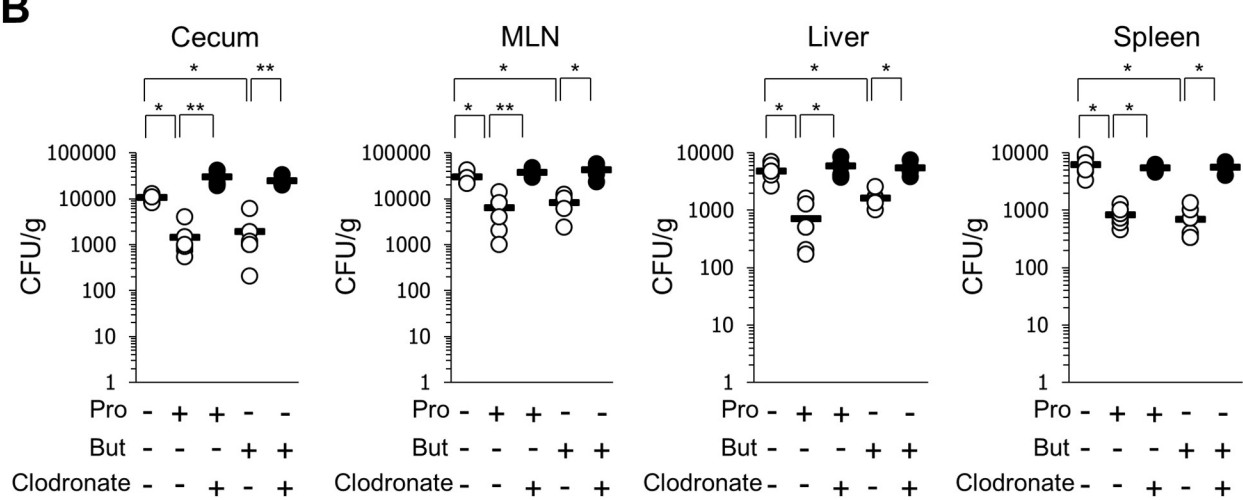

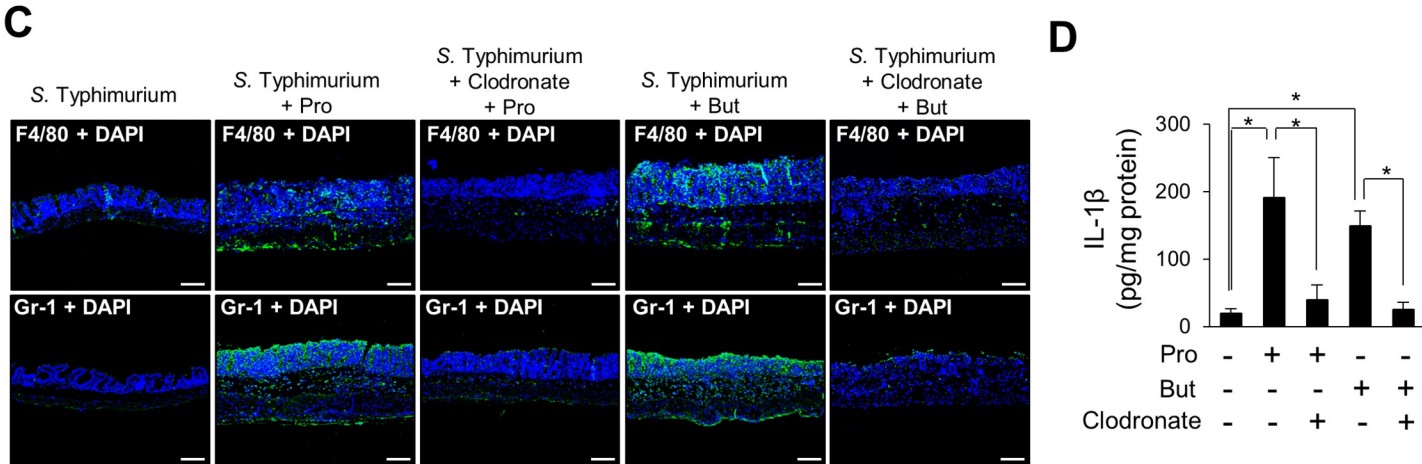

**Fig 6. SCFAs exert host-protective effects in a macrophage-dependent manner.** Mice were given antibiotics for 1 month prior to being given drinking water containing 300 mM propionate or 300 mM butyrate for 1 week. Then, clodronate liposomes (56 mg per kg of body weight) were intraperitoneally administered to the mice 24 hours prior to oral infection with *S.* Typhimurium ($10^8$ bacteria). (A) Effect of clodronate liposomes (macrophage-depleting agent) on the survival of propionate- or butyrate-administered wild-type mice infected with *S.* Typhimurium strain A. *P*-values were determined by the log-rank test (*n* = 6 per group). Data are listed in S1 Data. (B) Bacterial counts in cecum tissues, MLN, liver, and spleen were determined 2 days after infection. Cecum contents and each tissue were homogenized in PBS, and then the homogenates were plated on the *Salmonella Shigella* selection agar (Difco SS agar) and CFU were counted. Bars indicate the mean (*n* = 6 per group). *P < 0.05, **P < 0.01. Data are listed in S1 Data. (C) Immunostaining of cecum tissues after infection with *S.* Typhimurium strain A using anti-F4/80 (macrophage marker) and anti-Gr-1 (neutrophil marker) antibodies. Scale bars = 100 μm. (D) IL-1β production in the cecum as measured by ELISA. Data are the mean ± SD. *P < 0.05, ** P < 0.01. Data are listed in S1 Data. CFU, colony-forming unit; DAPI, 4',6-diamidino-2-phenylindole; IL, interleukin; MLN, mesenteric lymph node; *S.* Typhimurium, *S. enterica* serovar Typhimurium; SCFA, short-chain fatty acid; SD, standard deviation.

## Dietary fiber prevents *S.* Typhimurium infection through an ASC-dependent mechanism

Indigestible dietary fibers are fermented to SCFAs by gastrointestinal microorganisms and thus contribute to the SCFA concentration in the gut [39,40]. The effect of PHGG (mean molecular mass of 20 kDa [PHGG-R] or 14 kDa [PHGG-HG]), a water-soluble dietary fiber, on *S.* Typhimurium infection was examined. Administration of PHGG-R or PHGG-HG significantly increased the butyrate concentration in the cecum (Fig 7A). Administration of PHGG also prolonged the survival of *S.* Typhimurium–infected mice (Fig 7B). Enhanced resistance to *S.* Typhimurium infection induced by PHGG was not observed in *ASC*$^{-/-}$ mice, indicating that the protective effect of indigestible dietary fiber against the pathogen depends on ASC (Fig 7C). Administration of PHGG enhanced IL-1β production in *S.* Typhimurium–infected cecum tissues (Fig 7D). These results suggested that indigestible dietary fiber protects against *S.* Typhimurium infection by increasing butyrate production in the gut via an ASC-dependent mechanism.

SCFAs are difficult to visualize by conventional mass spectrometry with high-vacuum ionization system, because of their high volatility and low molecular mass. To overcome the technical difficulties, we utilized a high-resolution mass spectrometer using atmospheric-pressure ionization, which enabled us to stably detect SCFAs without sublimation. *S.* Typhimurium–infected cecal tissues lose structural stability owing to the inflammation, resulting in accidental cracks of the cecal mucosa and submucosa during the preparation of frozen sections. Despites this, the boundary between the cecal lumen and the mucosa could be identified in these sections and was indicated by a dotted line (Fig 7E). In SPF wild-type mice, normal cecal mucosal areas were abundant in citric acid, whereas only trace signals of propionate and butyrate were detected, indicating an intact mucosal barrier (Fig 7E left panel). In *S.* Typhimurium–infected SPF wild-type mice, marked accumulation of propionate and butyrate was observed in the inflamed cecal mucosa (Fig 7E middle panel). This regional distribution of SCFAs in the inflamed cecal mucosa suggested increased permeability of the damaged mucosa to SCFAs. Moreover, PHGG-R administration in *S.* Typhimurium–infected SPF wild-type mice enhanced regional access of propionate and butyrate into the inflamed cecal mucosa (Fig 7E right panel). These findings suggested that SCFAs produced from dietary fibers reach the nests of inflammatory cells infiltrating into the cecal mucosa and activate the inflammasomes of macrophages residing there.

## Discussion

The inflammasome is a multimeric protein complex that consists of NLRPs, ASC, and caspase-1. Its activation leads to the production of mature proinflammatory cytokines IL-1β and IL-18 [41]. Chemical and biological approaches using high-performance affinity nanobeads revealed that SCFAs bind to the PYRIN domain of ASC and NLRP3 and thereby promote the activation of the inflammasome, resulting in inflammatory cytokine production, both in vitro and in vivo.

The inflammasome activation is caused by the formation of inflammasome complex by sensing danger signals including damage-associated molecular patterns (DAMPs) or pathogen-associated molecular patterns (PAMPs) by NLRPs. SCFAs did not induce the activation of inflammasome in macrophages without *S.* Typhimurium infection, indicating that enhanced activation of the inflammasome by SCFAs requires the sensing the DAMPs or PAMPs by NLRPs (Fig 3B, 3C and 3D). SCFAs promoted the binding of NLRPs to ASC by binding to the ASC PYRIN domain, resulting in the enhanced activation of inflammasome in response to PAMPs (Figs 1 and 2). The interaction between the PYRIN domains of ASC and NLRPs is

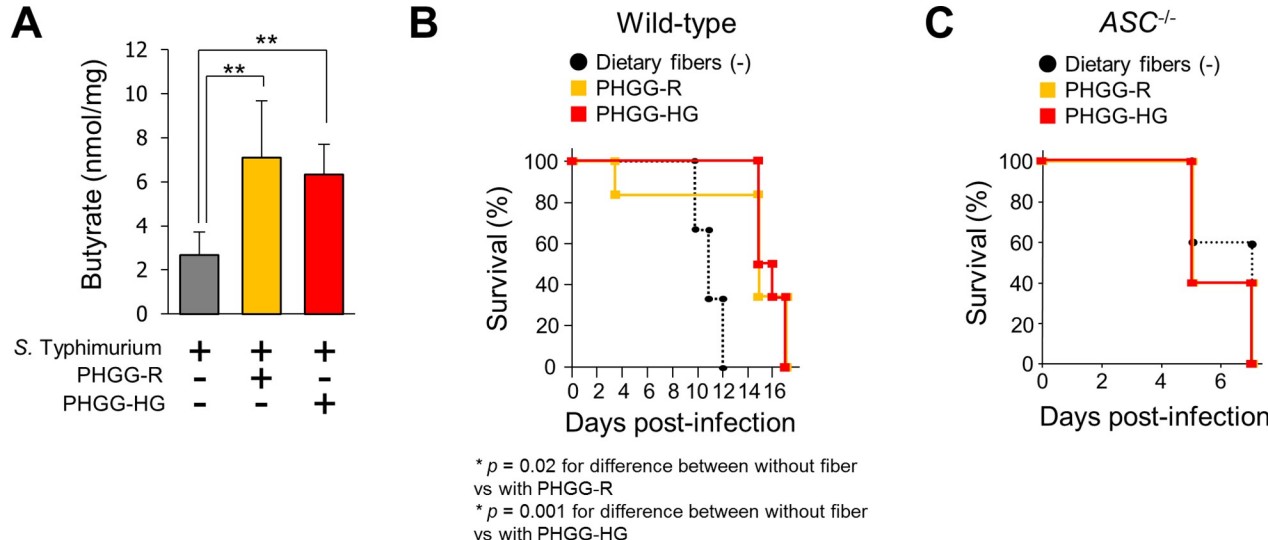

**Fig 7. Dietary fiber enhances host defense against *S.* Typhimurium infection.** (A) Dietary fiber (PHGG-R [mean molecular mass of 20 kDa] or PHGG-HG [mean molecular mass of 14 kDa]) was administered to SPF wild-type mice. Butyrate concentration in the cecal lumen as determined by LC-MS. Data are the mean ± SD (*n* = 5

per group). $**P < 0.01$. Data are listed in S1 Data. (B) Effect of each dietary fiber on the survival of wild-type infected with *S*. Typhimurium strain A ($n = 6$ per group). *P*-values were determined by the log-rank test. Data are listed in S1 Data. (C) Effect of each dietary fiber on the survival of $ASC^{-/-}$ mice infected with *S*. Typhimurium strain A ($n = 5$ per group). Data are listed in S1 Data. (D) IL-1β production in the cecum as determined by ELISA. Data are the mean ± SD. $*P < 0.05$. Data are listed in S1 Data. (E) Imaging mass spectrometric analysis of the localization of citrate, propionate, and butyrate/isobutyrate in cecum tissues from SPF wild-type mice (left panel), *S*. Typhimurium strain A–infected SPF wild-type mice (middle panel), and PHGGR-administrated *S*. Typhimurium strain A–infected SPF wild-type mice (right panel). Each tissue section was stained with HE (upper panel). Black dotted lines in imaging mass spectrometric images indicate a boundary between cecal lumen and cecal mucosa. Scale bars = 250 μm. ASC, apoptosis-associated speck-like protein; HE, hematoxylin–eosin; IL, interleukin; LC-MS, liquid chromatography–mass spectroscopy; PHGG, partially hydrolyzed guar gum; *S*. Typhimurium, *S. enterica* serovar Typhimurium; SD, standard deviation; SPF, specific pathogen-free.

required for activation of the inflammasome complex [20,21]. The PYRIN domain, which is composed of six helices, has distinct positively and negatively charged surface regions, and these bipolar surfaces are required for the binding between PYRIN domains [42]. Specifically, two basic residues within the ASC PYRIN domain (Lys21 and Arg41) are required for the interaction with the PYRIN domain of NLRPs [43]. We showed that SCFAs interact with the Lys-rich region of the ASC PYRIN domain, which includes the Lys21 residue, suggesting that SCFAs bind to the basic region of the PYRIN domain and alter electrostatic interactions, allowing it to more easily interact with other PYRIN domains. Lu and colleagues reported that the PYRIN and CARD domains of ASC are both filaments and that the PYRIN domain of ASC is nucleated by NLRP3, which senses PAMPs or DAMPs [44]. Although further investigation is needed to elucidate whether SCFAs affect the nucleation of the PYRIN domain of ASC filaments, our findings suggest that SCFAs promote the interaction of NLRPs with ASC by binding to the Lys-rich region of the ASC PYRIN domain, leading to enhanced inflammasome activation to protect against *S*. Typhimurium infection.

The Lys-rich region is highly conserved among PYRIN domain–containing proteins. Twenty-four PYRIN domain–containing proteins have been identified in humans to date, and they participate in apoptotic and inflammatory signaling pathways [42,45,46]. Further studies are necessary to unveil biological actions that result from SCFA binding to PYRIN domain–containing proteins other than the regulation of inflammasome activation as revealed in this study.

Although *S*. Typhimurium can metabolize butyrate by anaerobic β-oxidation because of its expression of the *ydiQRSTD* operon, which encodes metabolic enzymes related to β-oxidation of fatty acids [47], exposure to a high concentration of butyrate directly represses the expression of *S*. Typhimurium invasion genes (*invA* and *hilD*) [30,48,49]. Furthermore, Bronner and colleagues also showed that butyrate reduced *S*. Typhimurium colonization of Peyer's patches by inducing down-regulation of these invasion genes [49]. In our analyses, the IL-1β production in Peyer's patches was not enhanced by butyrate, suggesting that butyrate does not enhance inflammasome activation in Peyer's patches via an ASC-dependent mechanism (S15 Fig). These results support the idea that reduced *S*. Typhimurium colonization of Peyer's patches depends on the down-regulation of invasion gene expression by butyrate. On the other hand, in the cecum, our results show that butyrate enhanced IL-1β production via an ASC-dependent mechanism (Fig 5). Although further study is needed to elucidate the precise role of invasion genes in bacterial colonization, our findings indicate that butyrate enhances inflammasome activation to eliminate the bacteria in the cecum.

The SPI-1 of *S*. Typhimurium is known to induce the activation of the inflammasome, and therefore, in macrophages infected with *S*. Typhimurium SPI-1 deletion mutant strains, it has been reported that IL-1β production is small amounts and caspase-1 cleavage is undetectable [27,28]. Our results show that SCFAs enhanced LDH release and IL-1β production via an ASC-dependent mechanism in BMDMs infected with *S*. Typhimurium wild-type strain, leading to significantly less intracellular bacterial growth (Fig 3). On the other hand, SCFAs did not enhance the LDH release and IL-1β production in BMDMs without bacterial infection

(Fig 3). From these results, SCFAs were conceivable to promote the inflammasome activation induced by SPI-1 through mechanisms involving binding to ASC in *S*. Typhimurium–infected macrophages. Whereas SCFAs enhanced IL-1β production in BMDMs infected with SPI-1 knock-down *S*. Typhimurium through an ASC-dependent mechanism, the levels of IL-1β production were lower compared with that of BMDMs infected with *S*. Typhimurium wild-type strains (Figs 3D and S6D). From these findings, it was thought that SCFAs cannot enhance the pyroptosis contributing to the reduction of intracellular bacterial growth in BMDMs infected with *S*. Typhimurium SPI-1 knock-down strains.

Inflammasome activation by SCFAs enhanced the elimination of *S*. Typhimurium in macrophages, and administration of propionate and butyrate protected mice against *S*. Typhimurium infection through mechanisms involving binding of SCFAs to ASC. Based on these results, we suggest a potential mechanism by which dietary fibers or SCFAs activate the inflammasome complex, as shown in Fig 8. SCFAs enhance inflammasome activation by binding to the Lys-rich region of ASC, which leads to IL-1β production and pyroptosis in macrophages. Intracellular *S*. Typhimurium were eliminated from macrophages by pyroptosis, and IL-1β induced the recruitment of neutrophils into *S*. Typhimurium–infected cecal mucosa. Notably, inflammasome activation by SCFAs occurred independently of the canonical SCFA receptor GPR43. In line with these results, Franchi and colleagues reported that neutrophil infiltration was defective in IL-1 receptor–deficient mice infected with *S*. Typhimurium [50]. Furthermore, systemic infection by *S*. Typhimurium was enhanced in neutrophil-depleted mice [50,51]. We also showed that administration of SCFAs or dietary fibers robustly enhanced inflammatory cell infiltration in the cecal mucosa in response to *S*. Typhimurium infection and suppressed bacterial systemic invasion. Imaging mass spectrometric analysis revealed that SCFAs accumulate in the region of the cecal mucosa where inflammation is induced by *S*. Typhimurium infection. Collectively, the results of the present study provide evidence for a novel mechanism by which SCFAs activate innate immune responses in the intestinal mucosa to prevent severe systemic infection by *S*. Typhimurium.

Our results show that gut commensal bacteria-derived SCFAs enhance cecal inflammation in *S*. Typhimurium–infected mice in an ASC-dependent manner. The concentrations of propionate and butyrate in the cecal lumen of the SPF-C57BL/6 mice used in this study were the same as those of antibiotic-treated mice given exogenous SCFAs, which also induced inflammation (S9 Fig). Therefore, the localization of these SCFAs in the cecal lumen of SPF-C57BL/6 mice is thought to induce inflammation in the cecum in response to *S*. Typhimurium infection (Fig 5A). On the other hand, Barthel and colleagues reported that *S*. Typhimurium infection in SPF mice did not lead to cecal inflammation [52]. SCFAs are metabolites of gut commensal bacteria, and therefore, the concentration of SCFAs in the gut depends on the composition of the gut microbiota. Thus, the induction of cecal inflammation in SPF mice infected with *S*. Typhimurium may depend on the composition of the gut microbiota. The composition of gut commensal bacteria is reported to be significantly different in mice obtained from different vendors [53]. Although further study is needed to clarify the composition profile of gut microbiota associated with the induction of cecal inflammation in response to *S*. Typhimurium infection, our findings show that SCFAs induce cecal inflammation, which contributes to protection against bacterial infection through an ASC-dependent mechanism.

The administration of dietary fiber prior to bacterial infection significantly prolonged the survival of *S*. Typhimurium–infected mice by increasing the production of butyrate. In human epidemiologic studies, Africans reportedly were less susceptible to pathogenic bacterial infections than Europeans because of the high levels of intestinal SCFAs owing to high dietary fiber intake [54]. However, the precise molecular mechanisms of the protective effects of dietary fiber intake against pathogenic infection had not been clarified. Our finding that SCFAs

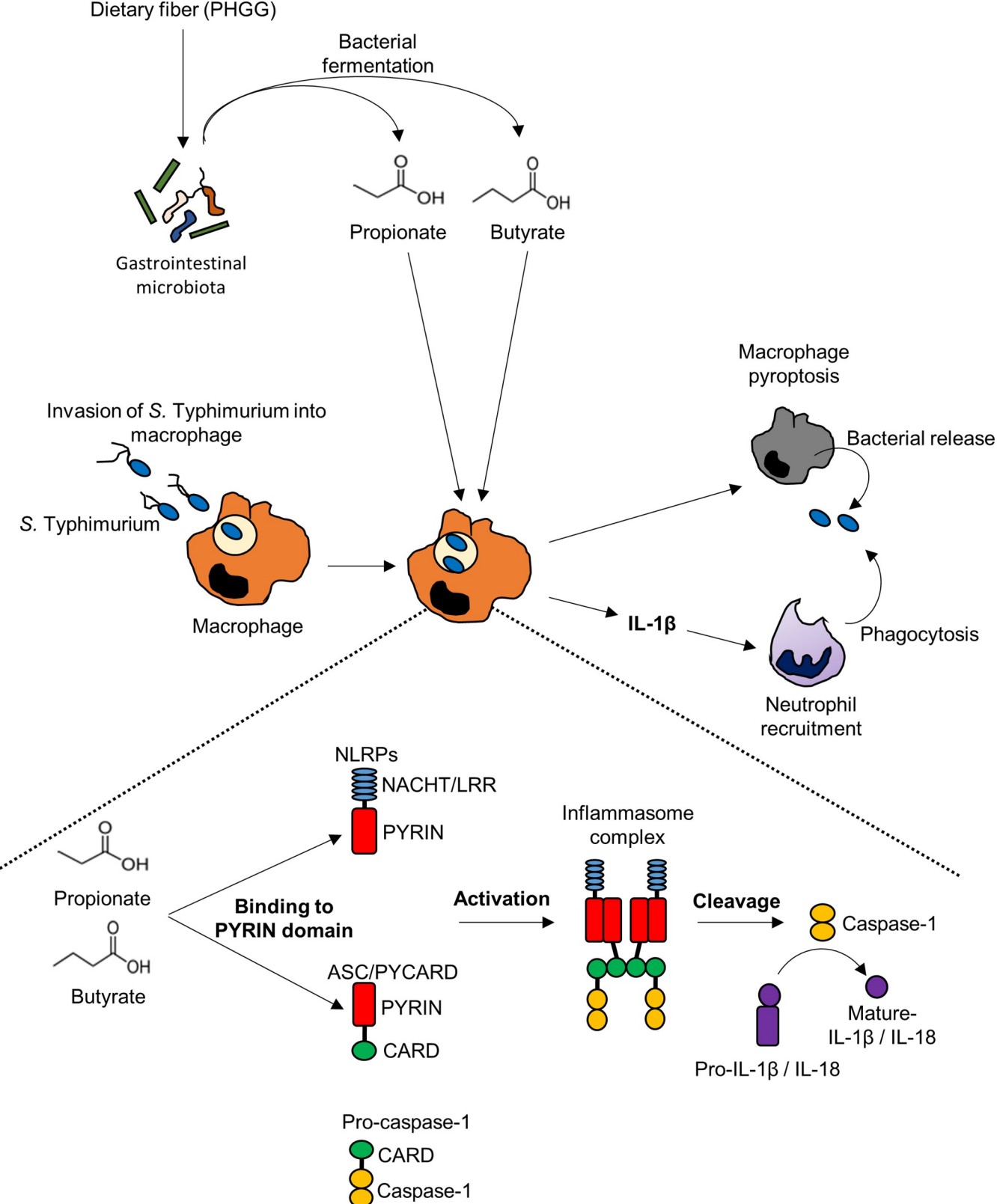

**Fig 8. Schematic diagram of SCFA-induced inflammasome activation and defense against *S.* Typhimurium.** SCFAs, which are abundantly produced by fermentation of dietary fibers in the gut, induce inflammasome activation in infected macrophages by enhancing oligomerization of the inflammasome

complex. Inflammasome activation contributes to the elimination of intracellular *S.* Typhimurium by pyroptosis. In addition, released pathogens are neutralized by infiltrating neutrophils, which are recruited to the inflamed tissue by IL-1β. ASC, apoptosis-associated speck-like protein; CARD, caspase activation and recruitment domain; IL, interleukin; LRR, leucine-rich repeat; NLRP, nucleotide-binding oligomerization domain-like receptor protein; PHGG, partially hydrolyzed guar gum; *S.* Typhimurium, *S. enterica* serovar Typhimurium; SCFA, short-chain fatty acid.

induced inflammasome activation supported these human epidemiological findings and provide evidence for the that fact dietary fiber intake can protect against gastrointestinal infections by activating innate immunity. Although further investigation is needed to grasp the precise mechanisms by which dietary fiber and SCFAs modulate intestinal infection, our findings provide new insights into potential therapeutic interventions to prevent pathogenic infections.

## Materials and methods

### Ethics statement

Animal care and use followed the regulations of the Act on Welfare and Management of Animals of Japan. All mice were bred and maintained under SPF conditions approved by Institutional Guidelines on Animal Experimentation at Keio University. All animal experiments were approved by the Keio University Animal Research Committee (no. 14016).

### Materials and antibodies

LPS (Calbiochem, San Diego, CA, United States of America, cat. no. 437627), ATP (Sigma-Aldrich, St. Louis, MO, USA, cat. no. A26209), nigericin (Cayman, Ann Arbor, MI, USA, cat. no. 28643-80-3), Imject Alum Adjuvant (Thermo Fisher Scientific, Waltham, MA, USA, cat. no. 77161), and anthrax lethal toxin (Calbiochem, cat. no. 176900) were used as inflammasome activators. Acetate (cat. no. 017–00256), propionate (cat. no. 163–04726), butyrate (cat. no. 023–05396), and lactate (cat. no. 128–00056) were purchased from Wako (Tokyo, Japan). The PHGG (PHGG, guar fiber, and galactomannan fiber) preparation (Sunfiber) used in this study was obtained from Taiyo Kagaku (Yokkaichi, Japan). The preparation was manufactured by treatment of guar gum with β-endogalactomannase from a strain of *Aspergillus niger*. The average molecular masses of PHGG-R and PHGG-HG were determined to be 20 and 14 kDa, respectively, by high-performance liquid chromatography. The following antibodies were used for western blotting: anti-caspase-1 (p20) (AdipoGen, San Diego, CA, USA, cat. no. AG-20B-0042; 1:500), anti-F4/80 antigen (M-300) (Santa Cruz Biotechnology, Santa Cruz, CA, USA, cat. no. sc-25830; 1:500), anti-ASC (N-15) (Santa Cruz Biotechnology, cat no. sc-22514; 1:500), anti-Gr-1/Ly-6G (R&D Systems, Minneapolis, MN, USA, cat. no. MAB1037; 1:500), anti-GPR43 (Y-15) (Santa Cruz Biotechnology, cat. no. sc-28420; 1:500), anti-NLRP3 (Abcam, Cambridge, UK, cat. no. ab214185; 1:1,000), and anti-β-actin (Sigma-Aldrich, cat. no. A2228; 1:1,000). Anti-mouse F4/80 antigen (A3-1) (Bio-Rad, Hercules, CA, USA, cat. no. MCA497; 1:50), anti-*Salmonella* Typhimurium LPS (Abcam, cat. no. ab8274250), and anti-Gr-1/Ly-6G (R&D Systems; 1:50) were used for immunohistochemistry.

### Cell and bacterial cultures

U937 cells (ATCC CRL-1593.2) were maintained in RPMI 1640 medium (Gibco/Thermo Fisher Scientific, cat. no. 11875093) supplemented with 10% fetal bovine serum (FBS). 293T cells (ATCC CRL-3216) were grown in Dulbecco's modified Eagle's medium (DMEM; Gibco, cat. no. 11965092) supplemented with 10% FBS. BMDMs were prepared by flushing bone marrow from femurs and tibiae with DMEM and were cultured in DMEM supplemented with 10% FBS and 20 ng/mL recombinant mouse macrophage-colony stimulating factor

(PeproTech, Rocky Hill, NJ, USA, cat. no. 315–02) for 7 days. *S*. Typhimurium strain A (obtained from the Research Institute for Microbial Diseases bacterial culture collection of Osaka University, Japan) [55] and *S*. Typhimurium ATCC14028S strain were cultured on Luria–Bertani (LB) agar (Becton-Dickinson, Franklin Lakes, NJ, USA, cat. no. 743–29229) overnight at 37˚C. *S*. Typhimurium strain A was used for in vitro analysis using BMDMs. *S*. Typhimurium strain A and ATCC14028S strain were used for in vivo analysis. Bacterial counts were determined by measuring the optical density of the bacterial suspension at 550 nm.

## Construction of the *S*. Typhimurium SPI-1 knock-down mutant strain

SPI-1 knock-down bacteria were constructed with an in-frame deletion of *sipB*, a transcriptional activator of SPI-1. An Flp recombination target (FRT)-flanked chloramphenicol-resistant marker cassette (KD3) fragment was amplified using the primers 5′-GTGTAGGCTGG AGCTGCTTC-3′ (forward) and 5′-ATGGGAATTAGCCATGGTCC-3′ (reverse) [56], and the PCR products were inserted into pGFPmut 3.1 (Clonetech, cat. no. 632370) downstream of the *gfp* gene. The *sipB*::*gfp*-KD3 fragment was amplified using the primers 5′-CGGAGACA GAGCAGCACAGTGAACAAGAAAAGGAATAATTATGCGTAAAGGAGAAGAACT-3′ (forward) and 5′-TACTAATTAACATATTTTTCTCCCTTTATTTTGGCAGTTTGTGTAGG CTGGAGCTGCTTC (reverse). The PCR products of the *sipB*::*gfp*-KD3 fragment were introduced into *S*. Typhimurium SR-11 χ3306/pKD46 by electroporation to construct the *sipB* in-frame replacement strain of *S*. Typhimurium SR-11 χ3306. The in-frame *sipB* deletion on the chromosome of *sipB*::*gfp*-KD3 fragment-electroporated *S*. Typhimurium SR-11 χ3306 was accomplished by λ Red-mediated recombination [57].

## Mice

All mice were bred and maintained under SPF conditions. All mice were fed the CE-2 diet (Japan Clea, Tokyo, Japan), which consists mainly of vegetable protein (soybean meal) and animal protein and contains a fiber source (wheat bran, defatted rice bran, and alfalfa meal). Male and female wild-type, $ASC^{-/-}$, and $GPRR43^{-/-}$ mice were used for experiments. To match the gender in the experiment, 6-week-old C57BL/6J male mice were purchased from SLC Japan (Shizuoka, Japan). $ASC^{-/-}$ mice were generated by replacing a 1.9-kb fragment encoding the *ASC* open reading frame with nLacZ and floxed pMC1 neo poly(A) [58]. To confirm the deletion of *ASC*, a 350-bp fragment of the *ASC* gene was amplified from genomic DNA using the primers 5′-CCAGGTAATGGTTAATCCCAGCAACC-3′ (forward) and 5′-TTGGCATTGCATGACATGGTGCACAC-3′ (reverse). $GPR43^{-/-}$ mice were generated by CRISPR-Cas9 gene editing. A guide RNA (gRNA) was designed for exon 3 of *GPR43* using the Optimized CRISPR Design tool (http://crispr.mit.edu/) (S4B Fig). The GPR43 gRNA was introduced into the pX330-U6-Chimeric_BB-CBh-hSpCas9 plasmid, and the gRNA expression vector was injected into the pronuclei of fertilized embryos collected from the oviducts of C57BL/6 mice. After injection, the zygotes were transferred into recipient females. To confirm the deletion of *GPR43*, a 390-bp fragment of the *GPR43* gene was amplified from genomic DNA using the primers 5′-CAGACTGGCACAGTTCCTTGATCC-3′ (forward) and 5′-GCG ATCACTCCATACAGTGGCC-3′ (reverse).

## 16S rRNA sequencing of microbiota and profile analysis

Fresh fecal samples were collected from five wild-type or $ASC^{-/-}$ mice, and genomic DNA was extracted using the NucleoSpin Microbial DNA Kit (TaKaRa, cat. no. U0235) according to the manufacturer's instructions. The extracted genomic DNA was sent to Macrogen (Tokyo, Japan) for metagenomics sequencing. The sequencing library was prepared by random

fragmentation of the DNA, followed by 5′ and 3′ adapter ligation using Herculase II fusion DNA polymerase (Agilent, cat. no. 600675) and Nextera XT Index Kit V2 (Illumina, cat. no. FC-131-1001), according Illumina's guidelines for 16S metagenomic sequencing library preparation (Illumina, Part #15044223 Rev. B). The amplicon of 16S rRNA was sequenced using paired-end sequencing on the Illumina MiSeq platform. The sequences were clustered de novo into operational taxonomic units (OTUs) using a CD-HIT-EST-based OTU analysis program (CD-HIT-OTU) and a cutoff of >97% similarity. The Shannon diversity index and the Inverse Simpson index, which account for both richness and evenness, and in which an increased value indicates greater diversity, were used to compare the microbial diversity of the OTUs in the libraries. The PCoA analysis of the weighted UniFrac distances was applied to the distance matrices for visualization of the microbial diversity of the OTUs in the libraries.

## Plasmid preparation

Full-length human *ASC* cDNA was amplified from a U937 cell cDNA library using the primers 5′-TTTTGAATTCATGGGGCGCGCGCGCGAC-3′ (forward) and 5′-TTTTGTCGACTCAGCTCCGCTCCAGGTCCTCCAC-3′ (reverse) and was introduced into the bacterial expression vector pGEX6P-1 (GE Healthcare, Waukesha, WI, USA, cat. no. 28954648). ASC deletion mutants (18 a.a. or 27 a.a.) were prepared using the primer 5′-TTTGAATTCGCCGAGGAGCTCAAGAAG-3′ for Δ18 and 5′-TTTGAATTCCTGTCGGTGCCGCTG-3′ for Δ27 and were introduced into pGEX6P-1. A FLAG-tagged ASC fragment was amplified using the primers 5′-TTTTGAATTCAATGGGGCGCGCGCGCGACGCC-3′ (forward) and 5′-TTTTAGATCTTCAGCTCCGCTCCAGGTCCTCCAC-3′ (reverse) and was introduced into pCMV10 (Sigma-Aldrich, cat. no. E7658). Human NLRP3 cDNA (OriGene, Rockville, MD, USA, cat. no. SC110747) was amplified using the primers 5′-TTTGGATCCATGAAGATGGCAAGCACCCGC-3′ (forward) and 5′-TTTCTCGAGCTACCAAGAAGGCTCAAAGAC-3′ (reverse) and was introduced into the mammalian expression vector pCDNA3.1 (Thermo Fisher Scientific, cat. no. V79020).

For the preparation of in vitro–translated proteins, full-length cDNAs of NLRP1, 3, 4, and 6, which were obtained from OriGene, were introduced into the SP6 promoter-containing vector pF3A WG Flexi (Promega, Madison, WI, USA, cat. no. L5671) using the following primers: NLRP1 forward, 5′-TTTTCGATACGCCATGGCTGGCGGAGCCTGGGGC-3′; NLRP1 reverse, 5′-TTTTGTTTAAACTCAGCTGCTGAGTGGCAGGAG-3′; NLRP3 forward, 5′-TTTTCGATACGCCATGAAGATGGCAAGCACCCGC-3′; NLRP3 reverse, 5′-TTTTGTTTAAACTTACTTCGGCTCATCTCTTTTTGCTTT-3′; NLRP4 forward, 5′-TTTTCGATACGCCATGGCAGCCTCTTTCTTCTCT-3′; NLRP4 reverse, 5′-TTTTGTTTAAACTCAGATCTCTACCCTTGTGAT-3′; NLRP6 forward, 5′-TTTTCGATACGCCATGGACCAGCCAGAGGCCCCC-3′; and NLRP6 reverse, 5′-TTTTGTTTAAACTCAGAAGGTCGAGATGAGTTCC-3′.

## Preparation of recombinant proteins

BL21 (DE3) cells were transformed with the pGEX-ASC expression vectors for full-length ASC or deletion mutants and were cultured in LB with ampicillin at 37˚C until an optical density at 600 nm (OD$_{600}$ nm) of 0.8 was reached. Protein expression was induced by the addition of 1 mM isopropyl-β-thiogalactopyranoside at 37˚C for 4 hours. Cell pellets were resuspended in buffer containing 20 mM Tris-HCl (pH 7.5), 100 mM NaCl, and 0.1% Tween-20, sonicated twice for 5 minutes at 4˚C, and centrifuged at 20,000*g* for 30 minutes. The supernatant was incubated with Glutathione Sepharose 4B (GE Healthcare, cat. no. 17075601) at 4˚C for 1 hour. Then, the resin was washed five times with the above buffer, the GST tag was cleaved by

the addition of Precision Protease (GE Healthcare, cat. no. 27084301), and the sample was incubated at 4°C for another 16 hours. Cleaved proteins were recovered from the supernatants and stored at −80°C.

For the preparation of in vitro–translated proteins, vectors expressing NLRP1, 3, 4, or 6 under the control of an SP6 promoter (5 μg) were incubated with 50 μL of TNT SP6 High-Yield Wheat Germ Protein Expression System (Promega, cat. no. L3260) and fluorescently labeled using FluoroTect Green$_{Lys}$ in vitro Translation Labeling System (Promega, cat. no. L5001) according to the manufacturer's instructions.

## Affinity purification using SCFA-conjugated affinity beads

For the preparation of SCFA-conjugated affinity beads, amino-modified affinity beads [16] were incubated with 1 mM succinate anhydride, malate anhydride, or glutarate anhydride in N,N-dimethylformamide at room temperature overnight. The conjugated beads were washed three times with N,N-dimethylformamide, and unreacted amine groups on the beads were masked by incubation with 5% acetic anhydride for 4 hours.

For the purification of SCFA-binding proteins, 0.2 mg of beads was equilibrated with binding buffer (20 mM HEPES-NaOH [pH 7.9], 100 mM NaCl, 10% glycerol, 1 mM EDTA, 0.1% NP-40) and incubated with 1 mg/mL U937 cell extract at 4°C for 1 hour. Bound proteins were eluted with sodium dodecyl sulfate (SDS) loading buffer, separated by SDS–polyacrylamide gel electrophoresis (PAGE), and visualized by silver staining (Wako, cat. no. 291–50301). Bound proteins were subjected to in-gel digestion with trypsin, and the peptide fragments were analyzed by ESI-MS (Hitachi, NanoFrontier, Tokyo, Japan).

For binding assays with recombinant ASC proteins, full-length ASC or deletion mutants (1 μg) were incubated with SCFA-conjugated beads, and binding was analyzed as described above. For competition assays, full-length ASC protein was preincubated with the indicated amount of propionate, butyrate, or lactate for 30 minutes, and then binding assays were performed.

## In vitro binding assay for NLRP3 and ASC

GST or GST-ASC (10 μg) was incubated with 10 μL of Glutathione Sepharose Beads (GE Healthcare) for 1 hour, and the beads were washed twice with PBS. The GST-ASC-immobilized resin was incubated with fluorescently labeled NLRP3 in 500 μL of binding buffer (20 mM HEPES-NaOH [pH 7.9], 100 mM NaCl, 10% glycerol, 1 mM EDTA, 0.1% NP40) with the indicated amounts of propionate, butyrate, or lactate for 1 hour. Bound proteins were washed three times with binding buffer and subjected to SDS-PAGE. Labeled NLRP3 protein was detected using a fluorescence imager (Typhoon FLA 9500; GE Healthcare), and GST or GST-ASC protein was visualized by Coomassie Brilliant Blue staining.

## Coimmunoprecipitation assays

FLAG-ASC (4 μg) and/or NLRP3 (8 μg) expression vectors were transfected into 293T cells using Lipofectamine 2000 (Invitrogen/Thermo Fisher Scientific, cat. no. 11668027) according to the manufacturer's instructions. The cells were incubated with the indicated amounts of propionate, butyrate, or lactate for 24 hours and then lysed with NP-40 lysis buffer (20 mM Tris-HCl [pH 7.5], 150 mM NaCl, 1% NP40). The lysates were incubated with 10 μL of equilibrated anti-FLAG (M2) agarose beads (Sigma-Aldrich, cat. no. A2220) at room temperature for 1 hour. Bound proteins were washed three times with 200 μl of binding buffer and eluted with 10 μL of 2 μg/mL FLAG peptide (Sigma-Aldrich, cat. no. F3290). The eluates were subjected to SDS-PAGE and visualized by western blotting using antibody against FLAG (Sigma-Aldrich, cat. no. F3040) or NLRP3 (Abcam).

## Cytokine measurement in U937 cells

U937 cells were seeded in 96-well plates and stimulated by the addition of 100 nM phorbol myristate acetate (Sigma-Aldrich, cat. no. P8139) for 24 hours. Then, the cells were treated with LPS (100 ng/mL) and ATP (5 μM), LPS and nigericin (5 μM), alum adjuvant (300 μg/mL), or anthrax lethal toxin (1 μg/mL) in the presence of the indicated concentrations of propionate, butyrate, or lactate for 12 hours. Cell culture supernatants were collected, and IL-1β (R&D Systems, cat. no. DLB50) and IL-18 (R&D Systems, cat. no. DL180) production was measured by ELISA according to the manufacturer's instructions.

## *S.* Typhimurium infection of BMDMs

BMDMs were seeded into 24- or 96-well plates at $2 \times 10^5$ or $5 \times 10^4$ cells per well. The cells were primed with 1 μg/mL LPS for 3 hours prior to *S.* Typhimurium strain A infection to induce pro-IL-1β expression. *S.* Typhimurium strain A cells were resuspended in DMEM. BMDMs were incubated with *S.* Typhimurium strain A at a multiplicity of infection of 5 for 10 minutes and then washed three times with PBS and incubated in DMEM containing 100 μg/mL gentamycin for 15 hours with or without SCFAs at a concentration of 1–10 mM. The cells were washed with PBS and lysed with PBS containing 1% Triton X-100. The cell lysates were plated on LB agar, and surviving intracellular *S.* Typhimurium strain A were enumerated as colony-forming units. IL-1β release in the cell culture supernatants was quantified using a mouse IL-1β ELISA kit (Abcam, ab100704) according to the manufacturer's instructions. LDH release was quantified using the CytoTox 96 cytotoxicity assay kit (Promega, G1780) according to the manufacturer's protocol. The percent LDH release was calculated as the ratio of the amount of LDH released under each experimental condition to the amount of LDH released by cells treated with 0.8% TritonX, as follows: % LDH release = [experimental LDH release ($OD_{490}$) − average LDH release in cells treated with culture medium ($OD_{490}$)]/[LDH release in cells treated with 0.8% TritonX-100 ($OD_{490}$) − average LDH release in cells treated with culture medium ($OD_{490}$)] × 100. Mature caspase-1 subunits are released into the supernatant in response to inflammasome activation [59]; therefore, to detect cleaved caspase-1 p20, cell culture supernatants were precipitated with 10% trichloroacetic acid on ice for 1 hour and then centrifuged at 10,000*g* at 4˚C for 30 minutes. Precipitated proteins eluted in SDS loading buffer were subjected to SDS-PAGE and visualized by western blotting using an anti-caspase-1 antibody (AdipoGen). BMDMs were incubated with the MCT inhibitor SR13800 (Calbiochem, 509663) for 24 hours prior to *S.* Typhimurium strain A infection.

## Quantitative reverse transcription–polymerase chain reaction (RT-qPCR)

*S.* Typhimurium strain A were cultured in the presence of 10 mM acetate, propionate, butyrate, or lactate in LB broth overnight at 37˚C, with agitation. Total RNA was isolated using the SV Total RNA Isolation System (Promega, Fitchburg, WI, USA, Z3100) and reverse-transcribed using the PrimeScript RT Reagent Kit (TaKaRa, Ohtsu, Japan, RR037A). PCRs were run using TB Green Premix Ex Taq II (TaKaRa, RR820S) in a Dice thermal cycler (TaKaRa, Ohtsu, Japan). The following primers were used: 16S rRNA mRNA, forward 5′-TGTTGTGGTTAATAACCGCA and reverse 5′-GACTACCAGGGTATCTAATCC; *fliB* mRNA, forward 5′-CTACGCGCTTCAGACAGATT and reverse 5′-GATCTGGGTGCGGTACAAA; *fliC* mRNA, forward 5′-GTAACGCTAACGACGGTATC and reverse 5′-ATTTCAGCCTGGATGGAGTC; *sipD* mRNA, forward 5′-TGAAAACGTTGTCGCAGTCT and reverse 5′-GCGCTGGAAATAAAACGGTA; *prgH* mRNA, forward 5′-AGATACGTTGTGGGCTCGTC and reverse 5′-TTCTTGCTCATCGTGTTTCG [60].

## *S.* Typhimurium infection of mice

All animal studies were approved by the Keio University Animal Research Committee (no. 14016). SPF wild-type mice were cohoused with SPF $ASC^{-/-}$ mice at a 1:1 gender ratio for 7 days prior to *S.* Typhimurium infection [61]. To clarify the role of SCFAs in the protection against *S.* Typhimurium infection, it was important to exclude the effects of endogenous SCFAs by completely eradicating gut commensal bacteria. Rakoff-Nahoum and colleagues reported that the combination therapy of four antibiotics (ampicillin, metronidazole, neomycin, and vancomycin) for mice was used for complete depletion of gut commensal bacteria [62]. Therefore, to exclude the effects of endogenous SCFAs derived from gut microbiota, 8- to 10-week-old mice were given drinking water containing ampicillin (1 g/L; Sigma-Aldrich, cat. no. A0166), metronidazole (1 g/L; Sigma-Aldrich, cat. no. M1547), neomycin (1 g/L; Sigma-Aldrich, cat. no. N1876), and vancomycin (0.5 g/L; Wako, cat. no. 222–01303) for 4 weeks prior to *S.* Typhimurium infection, as previously described [62,63]. Mice that lost more than 20% of their body weight by refusing to drink the antibiotic cocktail were excluded from experimental use, following the guidelines of the Keio University Animal Research Committee. The administration of antibiotics was stopped before bacterial infection. The mice were then given drinking water containing 300 mM SCFAs (propionate or butyrate) or 300 mM lactate for 1 week, according to a published protocol [64]. Since exogenous SCFAs were administered to antibiotic-treated mice in the drinking water, the concentrations of SCFAs in the cecal lumen were variable at the time of killing depending on the length of time since water intake. Therefore, the concentrations of SCFAs in the cecal lumen of mice were measured after an oral boost with 100 μL of 300 mM propionate or butyrate 2 hours prior to killing. *S.* Typhimurium ($10^8$ bacteria) were inoculated orally to the mice [65], and then survival was monitored daily. To assess the bacterial counts in the liver, spleen, and intestines, mice inoculated orally with *S.* Typhimurium ($2 \times 10^6$ bacteria) were humanely killed 2 days post infection, and the indicated tissues were collected and homogenized in PBS with Qiagen TissueLyser (Qiagen, Valencia, CA, USA). Serial dilutions of the homogenates were plated on the selection medium Difco SS agar (BD Pharmingen, cat. no. 274500) and colony-forming units were counted. For analysis of the effect of dietary fiber, PHGG (5 g per kg of body weight per day) was orally administered to mice for 3 weeks prior to *S.* Typhimurium infection, and survival was monitored. For analysis of the effect of macrophages, clodronate liposomes (FormuMax, Sunnyvale, CA, USA, F70101C-A) was intraperitoneally administered (56 mg per kg of body weight) to mice for 24 hours prior to *S.* Typhimurium infection to deplete macrophages. To determine sample size, a pilot study ($n = 6$) was performed to estimate the significant effect size of SCFAs, PHGG, or clodronate liposomes on *S.* Typhimurium infection in mouse. This estimated effect size ($n = 6$) gave a significant effect of SCFAs, PHGG, or clodronate liposomes on *S.* Typhimurium infection. This sample size was also used for no-statistical experiments.

## Histopathology of *S.* Typhimurium–infected mice

Cecum samples from mice infected with *S.* Typhimurium ($2 \times 10^6$ cells) for 2 days were collected, fixed in 4% paraformaldehyde overnight, and embedded in paraffin. Tissue sections (4 μm) were depleted of paraffin, rehydrated in a series of graded ethanol solutions, and stained with HE. HE-stained sections were observed using a NanoZoomer-XR (Hamamatsu Photonics, Tokyo, Japan). The inflammatory infiltrate score of HE-stained specimens were calculated based on the inflammatory infiltrate score defined by Koelink and colleagues' report [34]. For immunohistochemistry, tissue sections were subjected to antigen retrieval by heating for at 105˚C in Target Retrieval Solution (pH 9) (Dako, S2375) for 10 minutes. The tissue sections were then incubated with antibody against F4/80 antigen (Bio-Rad) or Gr-1 (R&D

Systems) at 4˚C overnight and then with 4',6-diamidino-2-phenylindole (DAPI; 1 μg/mL) and an Alexa Fluor 488–conjugated anti-rat or anti-goat IgG secondary antibody for 1 hour. Fluorescence images were obtained using an LSM710 confocal microscope (Carl Zeiss, Oberkochen, Germany). The staining quantifications of F4/80 and Gr-1 in the *S*. Typhimurium–infected cecum tissues were performed using ImageJ analysis software (National Institutes of Health, Bethesda, MD, USA). For the detection of IL-1β and Gr-1 in tissue, cecum samples were homogenized in RIPA buffer containing protease inhibitors. IL-1β was detected using a mouse IL-1β ELISA kit (Abcam, ab100704). Gr-1 expression was detected by western blotting using an anti-Gr-1 antibody (R&D Systems).

## Measurement of SCFAs

SCFAs were quantified by liquid chromatography–ESI-tandem MS (LC-ESI-MS/MS) with chemical derivatization [66,67]. Fecal samples (2–8 mg) were collected and suspended in 0.3 ml of EtOH with an internal standard (2-ethylbutyric acid). After centrifugation, SCFAs in the supernatant were derivatized with 2-nitrophenylhydrazine (2-NPH; at a final concentration of 10 mM), 1-ethyl-3-(3-dimethylaminopropyl) carbodiimide hydrochloride (10 mM), and 1.5% pyridine in 75% MeOH at room temperature for 30 minutes, with shaking. Aliquots were diluted 5-fold with 75% MeOH containing 0.5% formic acid and then subjected to LC-ESI-MS/MS. LC separation was conducted on a Mastro C18 column (2.1 mm I.D. × 150 mm L, 3 μm; Shimadzu, Kyoto, Japan) with a Nexera ultrahigh-performance liquid chromatography system (Shimadzu). The mobile phase consisted of 0.1% formic acid in water (A) and acetonitrile (B). The column oven temperature was maintained at 40˚C. The LC system was coupled with an LCMS-8060 triple-quadruple mass spectrometer (Shimadzu) operated in positive ion mode and multiple reaction monitoring mode with the following ion transitions: 2-NPH-derivatized acetate, *m/z* 196.3>43.1; derivatized propionate, *m/z* 210.1>57.2; derivatized butyrate and isobutyrate, *m/z* 224.2>71.1; derivatized valeric acid and isovaleric acid, *m/z* 238.1>85.1; 2-ethylbutyric acid (IS), *m/z* 252.2>99.2.

## Imaging mass spectrometry

Cecum samples from SPF wild-type mice, SPF wild-type mice infected with *S*. Typhimurium ($2 \times 10^6$ cells) for 2 days, and SPF wild-type mice administered PHGG-R (5 g per kg of body weight per day) for 3 weeks prior to *S*. Typhimurium infection were collected and embedded in the frozen embedding agent SCEM (Leica Microsystems, Tokyo Japan). Matrix-assisted laser desorption/ionization imaging analyses were performed as described previously [68]. Briefly, thin sections (8 μm) of the frozen-embedded caecum tissues were prepared with a cryo-microtome (CM3050; Leica Microsystems, Tokyo, Japan). The sections were attached onto indium tin oxide–coated glass slides (Bruker Daltonics GmbH, Leipzig, Germany) and were coated with 9-aminoacridine as the matrix (10 mg/mL, dissolved in 80% ethanol) by manual spraying with an airbrush (Procon Boy FWA Platinum; Mr. Hobby, Tokyo, Japan). The matrix was simultaneously applied to multiple sections in order to maintain consistent analyte extraction and co-crystallization conditions. Data were acquired using an orbitrap mass spectrometer (QExactive Focus; Thermo Fisher Scientific) coupled with an atmospheric-pressure scanning microprobe matrix-assisted laser desorption/ionization ion source (AP-SMALDI10; TransMIT GmbH, Giessen, Germany). The raster step size was set at 50 μm. For the orbitrap mass spectrometer, signals within a mass range of 50–192 were acquired with a mass resolving power of 70,000 at m/z 200, which enables calculation of the specific chemical formula of each ion signal. The spectral data were transformed to image data and analyzed using ImageQuest 1.0.1 (Thermo Fisher Scientific) and SCiLS 2019a (Bruker Daltonics) software.

## Statistical analysis

Data are presented as the mean ± standard deviation (SD). Means of multiple groups were compared by analysis of variance (ANOVA) followed by Tukey's tests using JSTAT statistical software (version 8.2). Animals were randomly assigned among the various groups. In repeated immunostaining experiments, conditions were randomized to account for potential ordering effects. All analyses were conducted with blinding to the experimental condition. Cumulative survival rates were analyzed by the Kaplan–Meier method, and differences in survival between subgroups were detected by log-rank testing using SPSS, version 22 for Windows (SPSS, Chicago, IL, USA). $P < 0.05$ was considered significant.

## Supporting information

**S1 Fig. SCFAs enhance IL-1β production induced by NLRP activators.** IL-1β production in U937 cells stimulated with SCFAs and LPS/nigericin, alum adjuvant, or lethal toxin as measured by ELISA. Data are the mean ± SD of three independent assays. One-way ANOVA analysis, $^*P < 0.05$, $^{**}P < 0.01$. Data are listed in S1 Data. ANOVA, analysis of variance; IL, interleukin; LPS, lipopolysaccharide; NLRP, nucleotide-binding oligomerization domain-like receptor protein; SCFA, short-chain fatty acid; SD, standard deviation.
(TIF)

**S2 Fig. Structures of fatty-acid derivatives, including mono-fatty acids, branched carboxylic acids, and di- and tri-carboxylic acids.**
(TIF)

**S3 Fig. Time-course analysis of inflammasome activation by SCFAs in *S.* Typhimurium–infected BMDMs.** (A) BMDMs derived from wild-type mice were infected with *S.* Typhimurium strain A at multiplicity of infection of 5 for 10 minutes and then incubated in DMEM containing 100 μg/mL gentamycin for the indicated time with or without treatment with 10 mM acetate (Ace), propionate (Pro), butyrate (But), or lactate (Lac). Bacterial cell numbers within BMDMs are shown. Data are the mean ± SD of three independent assays. One-way ANOVA analysis, $^*P < 0.05$ versus nontreated BMDMs (ctrl). Data are listed in S1 Data. (B and C) IL-1β production and LDH release in cell-culture media as determined by LDH assay and ELISA, respectively. Data are the mean ± SD of three independent assays. One-way ANOVA analysis, $^{**}P < 0.01$ versus nontreated BMDMs (ctrl). Data listed in S1 Data. ANOVA, analysis of variance; BMDM, bone marrow–derived macrophage; DMEM, Dulbecco's modified Eagle's medium; IL, interleukin; LDH, lactate dehydrogenase; ND, not detected (below the detection limit); *S.* Typhimurium, *S. enterica* serovar Typhimurium; SCFA, short-chain fatty acid; SD, standard deviation.
(TIF)

**S4 Fig. BMDMs derived from *ASC*- or *GPR43*-deficient mice.** (A) PCR analysis of *ASC* (350-bp PCR product) in genomic DNA isolated from wild-type or $ASC^{-/-}$ mice (left panel). The expression of ASC and F4/80, a macrophage marker, in the bone marrow or in BMDMs derived from wild-type or $ASC^{-/-}$ mice was analyzed by western blotting (right panel). (B) *GPR43*-deficient mice were generated using CRISPR-Cas9 gene editing. Cas9/gRNA-targeting sites in *GPR43*. Exons are indicated by closed boxes, and the boxed sequence begins 60 bp from the start codon and contains the targeting sequence in the coding region of exon 3. The gRNA-targeting sequence is underlined, and the PAM sequences are indicated in red. The CRISPR-Cas9 incision site is indicated by an arrow. (C) PCR analysis of GPR43 (390-bp PCR product) in genomic DNA isolated from wild-type or $GPR43^{-/-}$ mice (left panel). GPR43

expression in BMDMs derived from wild-type or *GPR43*$^{-/-}$ mice was analyzed by western blotting (right panel). ASC, apoptosis-associated speck-like protein; BMDM, bone marrow–derived macrophage; CRISPR-Cas9, clustered regularly interspersed short palindromic repeats–CRISPR-associated protein 9; gRNA, guide RNA; PAM, protospacer adjacent motif; PCR, polymerase chain reaction.
(TIF)

**S5 Fig. SCFAs do not directly affect *S*. Typhimurium.** (A) *S*. Typhimurium strain A was cultured in LB broth containing 10 mM SCFAs (acetate, propionate, butyrate, or lactate) for the indicated times at 37˚C, with agitation. Bacterial growth was monitored by measuring the optical density at 550 nm. Data are listed in S1 Data. (B) Effects of SCFAs on the expression of two flagellin genes of *S*. Typhimurium strain A. mRNA expression of *fliC* and *fliB* was measured by RT-qPCR. Data are the mean ± SD of three independent assays. Data are listed in S1 Data. LB, Luria–Bertani; RT-qPCR, quantitative reverse transcription–polymerase chain reaction; *S*. Typhimurium, *S. enterica* serovar Typhimurium; SCFA, short-chain fatty acid; SD, standard deviation.
(TIF)

**S6 Fig. Effect of SCFAs on inflammasome activation in macrophages infected with an *S*. Typhimurium SPI-1 knock-down mutant.** (A) Effects of SCFAs on the expression of SPI-1 genes (*sipD* and *prgH*) by *S*. Typhimurium. The expression of *sipD* and *prgH* in *S*. Typhimurium strain A and the *S*. Typhimurium SPI-1 knock-down strain were measured by RT-qPCR after SCFA exposure. Data are the mean ± SD of three independent assays. Student's *t* test, $^*P < 0.05$, $^{**}P < 0.01$. Data are listed in S1 Data. (B) BMDMs derived from wild-type or *ASC*$^{-/-}$ mice were infected with SPI-1 knock-down *S*. Typhimurium at a multiplicity of infection of 5 for 10 minutes and then were incubated in DMEM containing 100 μg/mL gentamycin for 15 hours with or without treatment with acetate (Ace), propionate (Pro), butyrate (But), or lactate (Lac). The percentages of surviving *S*. Typhimurium SPI-1 knock-down bacteria in SCFA-treated macrophages is shown relative to the survival in untreated macrophages (ctrl). Data are the mean ± SD of three independent assays. Data are listed in S1 Data. (C and D) BMDMs derived from wild-type or *ASC*$^{-/-}$ mice were infected with SPI-1 knock-down *S*. Typhimurium at a multiplicity of infection of 5 for 10 minutes and then were incubated in DMEM containing 100 μg/mL gentamycin for 15 hours with or without treatment with 10 mM acetate (Ace), 10 mM propionate (Pro), 10 mM butyrate (But), or 10 mM lactate (Lac). Cell supernatants were collected, and LDH release and IL-1β production were determined by LDH assay and ELISA, respectively. Data are the mean ± SD of three independent assays. One-way ANOVA analysis, $^*P < 0.05$. Data are listed in S1 Data. ANOVA, analysis of variance; BMDM, bone marrow–derived macrophage; DMEM, Dulbecco's modified Eagle's medium; LDH, lactate dehydrogenase; ND, not detected (below the detection limit); NS, not significant; RT-qPCR, quantitative reverse transcription–polymerase chain reaction; *S*. Typhimurium, *S. enterica* serovar Typhimurium; SCFA, short-chain fatty acid; SD, standard deviation; SPI-1, *Salmonella* pathogenicity island 1.
(TIF)

**S7 Fig. Pathogenicity of *S*. Typhimurium strain A and ATCC14028S strain against mice.** (A) Survival of SPF wild-type mice infected with *S*. Typhimurium strain A (10$^8$ bacteria) (black circle) and infected with *S*. Typhimurium ATCC14028S strain (10$^8$ bacteria) (blue circle). *n* = 6 per group. Data listed in S1 Data. (B) HE staining of cecum tissues from SPF wild-type infected with *S*. Typhimurium strain A or *S*. Typhimurium ATCC14028S strain. Scale bars = 50 μm. HE, hematoxylin–eosin; *S*. Typhimurium, *S. enterica* serovar Typhimurium;

SPF, specific pathogen-free.
(TIF)

**S8 Fig. Gut microbial diversity analysis of wild-type or *ASC*$^{-/-}$ mice.** (A) The Shannon diversity index and the Inverse Simpson index were analyzed to compare the microbial diversity between wild-type and *ASC*$^{-/-}$ mice. The box-and-whisker plots show the full range of variation, the interquartile ranges, and the median values. The data points indicate the diversity indices of each mouse. Data are listed in S1 Data. (B) The two principal coordinates from the PCoA of the weighted UniFrac distances were plotted to the distance matrices for visualization of the microbial diversity. The data points indicate each mouse. Data are listed in S1 Data. ASC, apoptosis-associated speck-like protein; PCoA, principal coordinate analysis.
(TIF)

**S9 Fig. Effect of antibiotics administration on the concentrations of propionate, butyrate, and lactate in the cecal lumen.** Mice were given drinking water containing ampicillin (1 g/L), metronidazole (1 g/L), neomycin (1 g/L), and vancomycin (0.5 g/L). After 4 weeks, the mice were humanely killed and cecum samples were collected. Antibiotic-treated mice were given drinking water containing 300 mM propionate, butyrate, or lactate for 1 week, and cecum samples were collected following a boost with propionate, butyrate, and lactate 2 hours prior to killing. $n = 5$ per group. One-way ANOVA analysis, $^{**}P < 0.01$. Data are listed in S1 Data. ND, not detected (below the detection limit).
(TIF)

**S10 Fig. Propionate and butyrate improve the survival of *GPR43*$^{-/-}$ mice infected with *S.* Typhimurium.** *GRP43*$^{-/-}$ mice treated or not with propionate ($n = 7$ mice per group) or butyrate ($n = 6$ mice per group) were infected orally with $10^8$ *S.* Typhimurium, and survival was monitored. *P*-values were determined by the log-rank test. Data are listed in S1 Data. *S.* Typhimurium, *S. enterica* serovar Typhimurium.
(TIF)

**S11 Fig. Effects of propionate, butyrate, or lactate on the *S.* Typhimurium–infected bacterial number in wild-type and *ASC*$^{-/-}$ mice.** (A and B) Bacterial cells in the cecum contents, cecum tissues, MLN, liver, and spleen were counted 2 days after infection. Cecum contents and each tissue were homogenized in PBS, and then the homogenates were plated on the *Salmonella Shigella* selection agar (Difco SS agar) and colony-forming units were counted. Bars indicate the mean ($n = 6$ per group). One-way ANOVA analysis, $^{*}P < 0.05$, $^{**}P < 0.01$. Data are listed in S1 Data. ANOVA, analysis of variance; ASC, apoptosis-associated speck-like protein; MLN, mesenteric lymph node; *S.* Typhimurium, *S. enterica* serovar Typhimurium.
(TIF)

**S12 Fig. The inflammatory infiltrate score and the staining intensity in *S.* Typhimurium–infected cecal mucosa and submucosa of wild-type or *ASC*$^{-/-}$ mice.** (A) The inflammatory infiltrate scores of HE-stained specimens were calculated. Bars indicate the mean ($n = 6$ per group). One-way ANOVA analysis, $^{*}P < 0.05$, $^{**}P < 0.01$. Data are listed in S1 Data. (B) The staining intensities for F4/80 and Gr-1 in the *S.* Typhimurium–infected cecum tissues of wild-type or *ASC*$^{-/-}$ mice were quantified using ImageJ analysis software. Bars indicate the mean ($n = 6$ per group). One-way ANOVA analysis, $^{*}P < 0.05$. Data are listed in S1 Data. ANOVA, analysis of variance; ASC, apoptosis-associated speck-like protein; HE, hematoxylin–eosin; *S.* Typhimurium, *S. enterica* serovar Typhimurium.
(TIF)

**S13 Fig. The effect of SCFA treatment on SPF mice without *S*. Typhimurium infection.** HE staining of cecum tissues from SCFAs-treated SPF wild-type or SPF $ASC^{-/-}$ mice without *S*. Typhimurium infection. Scale bars = 100 μm. ASC, apoptosis-associated speck-like protein; HE, hematoxylin–eosin; *S*. Typhimurium, *S. enterica* serovar Typhimurium; SCFA, short-chain fatty acid; SPF, specific pathogen-free.
(TIF)

**S14 Fig. Detection of *S*. Typhimurium in the cecal mucosa and submucosa of mice administered clodronate-encapsulated liposomes.** Immunostaining of cecum tissues using an anti-F4/80 antibody and an anti–*S*. Typhimurium LPS antibody. Scale bars = 50 μm. *S*. Typhimurium, *S. enterica* serovar Typhimurium.
(TIF)

**S15 Fig. SCFAs does not activate macrophages in MLN and Peyer's patches through an ASC-dependent mechanism.** (A) IL-1β production in MLN as determined by ELISA (*n* = 4 per group). (B) IL-1β production in Peyer's patches as determined by ELISA (*n* = 4 per group). Data are listed in S1 Data. ASC, apoptosis-associated speck-like protein; IL, interleukin; MLN, mesenteric lymph node; ND, not detected (below the detection limit).
(TIF)

**S1 Data. Excel spreadsheet with data listed for Fig 2C and 2D, Fig 3A, 3C, 3D, 3E and 3F, Fig 4A, 4C and 4D, Fig 5D, Fig 6A, 6B and 6D, Fig 7A, 7B, 7C and 7D, S1 Fig, S3 Fig, S5 Fig, S6 Fig, S7A Fig, S8 Fig, S9 Fig, S10 Fig, S11 Fig, S12 Fig and S15 Fig.**
(XLSX)

## Acknowledgments

We are grateful to the Collaborative Research Resources, Keio University School of Medicine, for technical assistance.

## Author Contributions

**Conceptualization:** Yasuaki Kabe, Hidekazu Suzuki, Makoto Suematsu.

**Data curation:** Hitoshi Tsugawa, Yasuaki Kabe, Yuki Sugiura, Keiyo Takubo.

**Formal analysis:** Hitoshi Tsugawa, Yasuaki Kabe, Ayaka Kanai, Yuki Sugiura, Shun'ichiro Taniguchi, Toshio Takahashi, Hidenori Matsui, Keiyo Takubo, Kenya Honda.

**Funding acquisition:** Yasuaki Kabe, Shigeaki Hida.

**Investigation:** Hitoshi Tsugawa, Yasuaki Kabe.

**Methodology:** Yasuaki Kabe, Hidenori Matsui, Hiroshi Handa.

**Resources:** Zenta Yasukawa, Hiroyuki Itou, Kenya Honda.

**Writing – original draft:** Yasuaki Kabe, Hiroshi Handa, Makoto Suematsu.

**Writing – review & editing:** Yasuaki Kabe.

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
