## [Editor Report · Decision Letter 0]

7 Feb 2020

Dear Dr Kabe, 

Thank you for submitting your manuscript entitled "Short-chain fatty acids bind to apoptosis-associated speck-like protein to activate inflammasome complex to prevent Salmonella infection" for consideration as a Research Article by PLOS Biology.

Your manuscript has now been evaluated by the PLOS Biology editorial staff as well as by an academic editor with relevant expertise and I am writing to let you know that we would like to send your submission out for external peer review.

Please re-submit your manuscript within two working days, i.e. by Feb 09 2020 11:59PM.

Kind regards,

Lauren A Richardson, Ph.D

Senior Editor

PLOS Biology

---

## [Decision Letter · Decision Letter 1]

16 Mar 2020

Dear Dr Kabe,

Thank you very much for submitting your manuscript "Short-chain fatty acids bind to apoptosis-associated speck-like protein to activate inflammasome complex to prevent Salmonella infection" for consideration as a Research Article at PLOS Biology. Your manuscript has been evaluated by the PLOS Biology editors, an Academic Editor with relevant expertise, and by two independent reviewers.

In light of the reviews (below), we will not be able to accept the current version of the manuscript, but we would welcome re-submission of a much-revised version that addresses all of the reviewers' concerns. We cannot make any decision about publication until we have seen the revised manuscript and your response to the reviewers' comments. Your revised manuscript is also likely to be sent for further evaluation by the reviewers.

We expect to receive your revised manuscript within 2 months. 

**IMPORTANT - SUBMITTING YOUR REVISION**

*Re-submission Checklist*

*Published Peer Review*

*PLOS Data Policy*

*Blot and Gel Data Policy*

Sincerely,

Di Jiang, PhD

Senior Editor

PLOS Biology

REVIEWS:

Reviewer #1: Short chain fatty acids have been of continued interest due to their various effects on the microbiota and the host. The presented study by Tsugawa et al. sheds light on the mechanism of how SCFAs modulate the immune response and prolong survival after Salmonella infection. The authors determined that SCFAs bind to a lysine-rich domain of ASC, which facilitates binding to NLRPs. Macrophages exposed to SCFAs killed Salmonella more efficiently in vitro. In vivo, SCFA administration to antibiotics-treated mice increased their survival. No increased survival was seen for SCFA administration to ASC-/- mice. Administration of soluble fiber increased survival of Salmonella-infected mice. 

This manuscript uncovers an interesting mechanism that should be of interest to a broad audience, as SCFA-mediated enhancement of the immune response will likely occur also with other intestinal pathogens. The experiments were in general performed rigorously and include important controls. The authors used a variety of different methods, including imaging mass spectrometry, to support their hypothesis. However, for some experimental assays, additional information and clarifications need to be provided. 

The manuscript has two main weaknesses: 1) co-housing instead of using littermates and the lack of microbiota analysis and 2) no experimentation to exclude that added SCFAs regulate virulence gene expression in Salmonella. 

Co-housing apparently rendered the observed phenotype in Fig. 1 less pronounced. The microbiota analysis would, especially as fiber fermentation is subject of analysis, be a valuable addition to the paper. It could help dispel concerns that phenotypes are due to microbiota difference not resolved during co-housing. A number of studies have shown that SCFAs modulate Salmonella virulence gene expression. This was not addressed experimentally or discussed. 

The authors should address the following points:

1) Include analysis of the microbiota to show that WT and ASC-/- mouse microbiota is highly similar before the start of the experiment and during recovery. If there are differences before, differences in the microbiota that recovers after the stop of antibiotic treatment could be partially responsible for the observed phenotype. 

2) Related to point 1: The experimental setup in 4B is unclear. Please indicate with horizontal bars the periods when the antibiotic cocktail was administered and when SCFAs were administered. Was administration of the cocktail stopped before Salmonella infection or was a resistant Salmonella strain used? 

3) The authors did not indicate for how long they co-housed the mice. Extensive co-housing is accepted by researchers as a way to normalize the microbiota of strains of different origins. However, the authors used male mice for their experiments, where co-housing would presumably result in severe aggression. Please indicate for how long mice were co-housed and how they were housed during experiments (single vs. group). 

4) SCFAs like butyrate and propionate are known to down-regulate Salmonella SPI-1 expression at higher concentrations. Experiments with ASC-/- mice and BMDM seem to indicate that this is not a relevant mechanism in this model. No actual numbers, just percentages are given, so it is difficult to interpret. A probably relevant control would be an experiment with a SPI-1 deficient Salmonella strain and WT and ASC-/- BDMDs. 

5) Are the pictures in Fig. 5B truly representative? The immune cell recruitment with SCFAs is massive. Fig. 5C shows more moderate recruitment. Please provide pathology scores to assess all and not just representative mice. 

6) Were all experiments done exclusively with male mice? The purchased C57BL/6 mice were male, but no indication for other strains is given. Ideally (and most often now requested) experiments should be performed with male and female mice to exclude gender bias. 

Minor comments:

Manuscript is well written, but it needs to be thoroughly checked (e.g. "% of survive" or "infection to wild-type mice" and "scarify" in Fig. 4).

Multiple mouse models exist for Salmonella Typhimurium. Please provide the rationale for the choice of the mouse model you use (e.g. lower colonization and cecum inflammation than streptomycin pre-treatment mouse model). 

M&M clarifications needed:

* Fiber and composition of the diet is important for the observed phenotype. Please indicate which diet the mice were fed. 

* Mice are known to avoid drinking the antibiotic cocktail. How much weight did they lose before the beginning of the experiment? This should be noted to orient the reader to the fact that there are more differences between SPF and antibiotics-treated mice than the presence or absence of bacteria.

* The Salmonella strain used is likely ATCC 14028S and not ATCC 14208S?

* Rationale for using U937 cells, mention also in results. 

* Some key information, e.g. how long BMDMs were incubated with Salmonella, is only available in the figure legend and not in the M&M. Please add where appropriate. 

Fig. 3: Percent of LDH release is compared to which positive control?

Fig. 7: Imaging mass spectrometry is a very useful tool to illustrate the penetration of SCFAs into the tissue. In uninfected mice, the cecal tissue is easily identified, and also in sections of the infected mice. However, there seems to be absolutely no structure in the remaining parts, rather only huge rips. Are these artefacts from processing? Please explain this a bit more.

Fig. 3E: Why are here CFU/ml and not CFU as percent of control? How did this assay differ from the other similar assays in Fig. 3?

Fig. 3D: Please indicated that levels were below limit of detection and are not missing data

Fig. S4E: I assume you have done multiple tests to confirm the deletion, but this WB does not look convincing.

Fig. S7: Why was the extra SCFA boost necessary? Lines 252-254 seems to indicate that SCFA administration in the drinking water increases SCFA levels to similar levels as in SPF mice. However, the authors seem to measure an undefined boost and not what is available during Salmonella infection. 

Fig. S9: Bacterial numbers recovered from mice usually have log-normal distribution and are best displayed on a log scale.

Reviewer #2: Tsugawa et al provide evidence for an interesting model in which short chain fatty acids (SCFAs), which are produced through fermentation of dietary fiber by members of the gut microbiota, facilitate the interaction between the inflammasome pattern recognition receptor Nlrp3 and its adaptor Asc, leading to enhanced inflammasome activation. The authors pinpoint a lysine rich region of the pyrin domain as important for SCFA binding. Furthermore, SCFA treatment of macrophages leads to pyroptosis and restricts Salmonella intracellular survival. In addition, SCFA administration to mice induces inflammation and enhances resistance against Salmonella infection. These results suggest the intriguing possibility that eating a diet rich in fiber may protect against Salmonella infection. However, the manuscript is lacking in sufficient context with respect to previous studies involving Salmonella and SFCAs. A number of references important to this work are omitted and not discussed. 

Major comments:

1) There is no reference to PMID: 29447698 entitled "Genetic ablation of butyrate utilization attenuates gastrointestinal Salmonella disease". Among other things relevant to this manuscript, this paper showed that butyrate decreased Salmonella colonization of Peyer's patches in gnotobiotic mice.

2) If SCFAs induce ROS by neutrophils (pg 13 line 100), why would Salmonella colonization of the cecum decrease (Fig S9), as ROS induction by Salmonella itself leads to generation of alternative TEAs that allows Salmonella to outcompete microbiota (PMID: 27078066)?

3) What is the rationale behind using STm strain A and needing to compare it to ATCC 14208S?

4) An explanation of how Nlrp3 normally gets activated and interacts with Asc is lacking. What's known and how could SCFAs fit in? Would SCFAs affect nucleation of Asc filaments by Nlpr3, as in Lu et al 2014 Cell (PMID: 24630722).

5) Is it possible that the top band in Fig 1B found in the proprionate and butyrate samples but not the acrylate sample is polymerized Asc?

6) Fig 3A: What is shown in the images? If they are wells with infected monolayers, why are the cells growing in clumps? A scale bar would be useful.

7) Why was procasp-1 and casp-1 precipitated out of the infected monolayer supernatant using acid rather than probed from cell lysate (Fig 3B)? 

8) Expression of SPI-1 genes in Salmonella were previously shown to modulate SPI-1 gene expression (PMID: 19433544, PMID: 12453229), but this is not mentioned in the manuscript nor are these papers referenced. Since the SPI-1 T3SS induces inflammasome activation, this is important to address.

9) Fig S3A: Why are Salmonella killed even in the absence of SCFAs during intracellular residence even at time points where no pyroptosis is detected?

10) Are the mice in Fig 4C and Fig 6 pre-treated with antibiotics?

11) In Barthel et al 2003 I&I (PMID: 12704158), Salmonella infection in the absence of antibiotic pre-treatment did not lead to cecal inflammation, which is at odds with Fig 5A.

12) Does SCFA treatment of naïve mice alter tissue architecture?

13) Where are the Salmonella replicating in clodronate-treated mice? Inside other cell types? Extracellularly?

Minor comments:

1) Needs to be made clear that bacteria do not make SCFAs only to regulate immune responses (pg. 11 lines 62-3).

2) Grammatical errors are found in the manuscript. For example, pg 11 line 64.

3) Pg 13 line 107, should be SCFA-bound, not SCFA-found.

4) Fig 4B, "scarify" would be "sacrifice".

---

## [Decision Letter · Decision Letter 2]

15 Jun 2020

Dear Dr Kabe,

Thank you for submitting your revised Research Article entitled "Short-chain fatty acids bind to apoptosis-associated speck-like protein to activate inflammasome complex to prevent Salmonella infection" for publication in PLOS Biology. I have now obtained advice from the original reviewers and have discussed their comments with the Academic Editor. 

Based on the reviews, we will probably accept this manuscript for publication, assuming that you will modify the manuscript to address the remaining points raised by the reviewers. Please also make sure to address the data and other policy-related requests noted at the end of this email.

We expect to receive your revised manuscript within two weeks. Your revisions should address the specific points made by each reviewer. In addition to the remaining revisions and before we will be able to formally accept your manuscript and consider it "in press", we also need to ensure that your article conforms to our guidelines. A member of our team will be in touch shortly with a set of requests. As we can't proceed until these requirements are met, your swift response will help prevent delays to publication.

*Copyediting*

*Published Peer Review History*

*Early Version*

*Submitting Your Revision*

Sincerely,

Di Jiang, PhD

PLOS Biology

ETHICS STATEMENT:

-- Please create a separate subsection entitled "Ethics Statement" and place it in the beginning of the Methods section, and please include all relevant information described below. 

-- Please include the full name of the IACUC/ethics committee that reviewed and approved the animal care and use protocol/permit/project license. Please also include an approval number.

-- Please include the specific national or international regulations/guidelines to which your animal care and use protocol adhered. Please note that institutional or accreditation organization guidelines (such as AAALAC) do not meet this requirement.

-- Please include information about the form of consent (written/oral) given for research involving human participants. All research involving human participants must have been approved by the authors' Institutional Review Board (IRB) or an equivalent committee, and all clinical investigation must have been conducted according to the principles expressed in the Declaration of Helsinki.

DATA POLICY:

Regardless of the method selected, please ensure that you provide the individual numerical values that underlie the summary data displayed in the following figure panels as they are essential for readers to assess your analysis and to reproduce it: Figures 2CD, 3ACDEF, 4ACD, 5DE, 6ABD, 7A-D, S1, S3A-C, S5AB, S6A-D, S7A, S8, S9, S10, S11, S12AB, S15. NOTE: the numerical data provided should include all replicates AND the way in which the plotted mean and errors were derived (it should not present only the mean/average values).

Reviewer remarks:

Reviewer #1: The authors very carefully and thoughtfully revised their manuscript. They added a large amount of additional experimental data to support their hypothesis and address the reviewer's concerns. The additional clarifications helped to make the manuscript more accessible to a larger readership. I have only two minor comments. 

Comments:

The authors added a microbiome analysis as requested. Unfortunately, the graphs that were chosen for the manuscript do not actually show that there is no difference in the microbiome composition. The graphs show that there is no difference in alpha diversity, which measures the number and abundance of taxa in a given sample. Alpha diversity can be exactly the same in individuals of two groups while the community composition (which taxa are present) between the groups can be completely different. In order to show that there are no significant differences in community structure between the experimental groups, beta diversity needs to be shown. The microbiota analysis is clearly not the focus of the paper or the expertise of the authors. However, the authors went through the trouble of adding this important piece of data for their mouse experiments. It should therefore be actually significant data that supports their hypothesis. The microbiome data are already available, so adding beta diversity PCA plots should be a minor modification. 

The description of the "boost" is misleading. In the results section, the concentrations of the SCFAs in water are not given, only the concentrations for the "boost". This gives the impression that the concentration given by oral gavage is higher than what the mice are drinking throughout the experiment. Through the clarification in M&M I learned that this is not the fact, that the concentration in the boost and in the water are the same. As you explained in the reply to my comment, this was more a "forced drinking" to prevent differences in measurements resulting from that once mouse had recently drank and another an hour ago. You might want to clarify this in the results section. 

Reviewer #2: 1) Why is intracellular growth data presented as % CFU in Fig 3 but as CFU/mL in Fig S6B? Also, line 257 should read "non-significant effect" rather than "weak effect".

2) The SPI-1 knockdown mutant induced ASC-dependent IL-1� in BMDMs synergistically with SCFAs, but SCFAs did not potentiate LDH release nor diminish intracellular growth of the SPI-1 mutant. This is in contrast to WT bacterial infection during SCFA treatment, where SCFAs led to significantly less intracellular growth of WT Salmonella and more LDH and IL-1� release. This result is not sufficiently explained.

1) "Butyrate" is misspelled in line 143. 

2) Grammatical errors persist.

---

## [Editor Report · Decision Letter 3]

24 Aug 2020

Dear Dr Kabe,

On behalf of my colleagues and the Academic Editor, Matthew K. Waldor, I am pleased to inform you that we will be delighted to publish your Research Article in PLOS Biology. 

Early Version

PRESS 

Thank you again for submitting your manuscript to PLOS Biology and for your support of Open Access publishing. Please do not hesitate to contact us if we can provide any assistance during the production process.

Kind regards,

Pamela Berkman

Publishing Editor, 

PLOS Biology

on behalf of

Gabriel Gasque 

Senior Editor

PLOS Biology